# Targeting the transcription factor HES1 by L-menthol restores protein phosphatase 6 in keratinocytes in models of psoriasis

Zhikai Wang[1], Yang Sun[1], Fangzhou Lou[1], Jing Bai[1], Hong Zhou[1], Xiaojie Cai[1], Libo Sun [1], Qianqian Yin[1], Sibei Tang[1], Yue Wu[1], Li Fan[1], Zhenyao Xu[1], Hong Wang[1], Xiaoyu Hu [2] & Honglin Wang [1] ✉

Protein Phosphatase 6 down-regulation in keratinocytes is a pivotal event that amplifies the inflammatory circuits in psoriasis, indicating that restoration of protein phosphatase 6 can be a rational strategy for psoriasis treatment. Through the phenotypic screen, we here identify L-menthol that ameliorates psoriasis-like skin inflammation by increasing protein phosphatase 6 in keratinocytes. Target identification approaches reveal an indispensable role for the transcription factor hairy and enhancer of split 1 in governing the protein phosphatase 6-upregulating function of L-menthol in keratinocytes. The transcription factor hairy and enhancer of split 1 is diminished in the epidermis of psoriasis patients and imiquimod-induced mouse model, while L-menthol upregulates the transcription factor hairy and enhancer of split 1 by preventing its proteasomal degradation. Mechanistically, the transcription factor hairy and enhancer of split 1 transcriptionally activates the expression of immunoglobulin-binding protein 1 which promotes protein phosphatase 6 expression and inhibits its ubiquitination. Collectively, we discover a therapeutic compound, L-menthol, for psoriasis, and uncover the dysfunctional the transcription factor hairy and enhancer of split 1- immunoglobulin-binding protein 1- protein phosphatase 6 axis that contributes to psoriasis pathology by using L-menthol as a probe.

Psoriasis is a chronic inflammatory disease affecting 2–3% of the global population[1] and associated with genetic predisposition[2], immune deregulation[3], and multiple environmental stimuli[4]. The abnormal interaction between hyperproliferative epidermal keratinocytes and self-reactive immune cells is a hallmark of psoriasis[5]. However, the underlying mechanisms of aberrant keratinocyte behaviors in psoriasis are incompletely understood.

Although current treatments provide multiple therapeutic options[6–9], there is no complete cure for psoriasis[10,11]. Further study is urged to explore new druggable targets which benefit treatment

outcomes for psoriasis, and natural compounds with biological activities are good tools to identify new druggable targets[12].

Protein phosphatase 6 (PP6) is an evolutionarily conserved and ubiquitously expressed serine/threonine-protein phosphatase known to regulate cell cycle progression[13], inflammatory signaling[14], and DNA damage repair[15]. PP6 is also crucial for skin homeostasis[16,17]. We previously reported that PP6 is diminished in the epidermis of psoriasis and imiquimod (IMQ)-induced mouse model. Epidermis-specific Pp6-deficient mice show histological, clinical, and gene expression changes consistent with the hallmarks of psoriasis[18,19]. As such, PP6

[1]Shanghai Institute of Immunology, Precision Research Center for Refractory Diseases, Shanghai General Hospital, Key Laboratory of Cell Differentiation and Apoptosis of Chinese Ministry of Education, Shanghai Jiao Tong University School of Medicine, Shanghai 200025, China. [2]Institute for Immunology and School of Medicine, Tsinghua Universiity, Beijing, China. ✉e-mail: honglin.wang@sjtu.edu.cn

downregulation in keratinocytes is a pivotal event that amplifies the inflammatory circuits in psoriasis, indicating that PP6-upregulating compounds may harbor the therapeutic potential for psoriasis.

Here, by performing phenotypic screening, we found that L-menthol alleviated psoriasis-like skin inflammation through upregulating PP6 in keratinocytes. With drug affinity responsive target stability assay (DARTS), the transcription factor hairy and enhancer of split 1 (HES1) was identified as the direct target of L-menthol in keratinocytes, and was protected from degradation in the presence of L-menthol. The upregulation of HES1 restored PP6 expression in inflamed keratinocytes and mitigated psoriasis-like skin inflammation. Mechanistically, HES1 promoted the transcription of IGBP1 which decreased the PP6-interacted E3 ubiquitin ligase, CHIP, and inhibited PP6 ubiquitination. Thus, we identified HES1 as a druggable target for psoriasis by using L-menthol as a molecular probe.

## Results

### L-menthol relieves skin inflammation via upregulating PP6

PP6 was dramatically decreased in the epidermis of psoriasis patients compared with that of healthy controls (Supplementary Fig. 1a). Besides, the increased epidermal thickness was observed in the 3D human skin equivalent cultured with keratinocytes knocked down of *PP6* gene (Supplementary Fig. 1b–f), indicating that PP6 deficiency in keratinocytes promotes epidermal hyperplasia in psoriasis. To screen for natural compounds that upregulate PP6 expression, we treated HaCaT cells with natural compounds from the Tsbiochem library in the presence of resiquimod (R848) (Supplementary Fig. 2a, b). Interestingly, we found that 10 μM L-menthol significantly increased the protein level of PP6 in HaCaT cells without affecting its mRNA level and cell viability (Fig. 1a, b and Supplementary Fig. 2c–e). L-Menthol is an effective natural compound widely used for counterirritant[20], but its function in psoriasis remained to be explored. To evaluate its effect on psoriasis-like skin inflammation, L-menthol was dissolved in ethanol (EtOH) and topically applied to IMQ-induced skin lesions (Fig. 1c). As expected, L-menthol alleviated inflammatory phenotypes (erythema, thickness, and scaling) (Fig. 1d and Supplementary Fig. 2f, g) and dramatically decreased the ear thickness (Fig. 1e) compared with the vehicle control. Histological examination of the skin lesions (Fig. 1f) showed that the epidermal thickness (acanthosis) (Fig. 1g) and dermal cellular infiltrate (Fig. 1h) in skin lesions were reduced significantly in mice treated with L-menthol compared with vehicle-treated controls. Meanwhile, L-menthol significantly reduced epidermal Ki67+ cell counts (Fig. 1i, j) and inflammatory cytokines (Fig. 1k) in skin lesions. The protein level of PP6 in the epidermis was dramatically upregulated by the topical application of L-menthol. (Fig. 1l and Supplementary Fig. 2h). Besides, L-menthol decreased the phosphorylation of C/EBP-β and subsequent arginase-1 production in inflamed keratinocytes (Supplementary Fig. 2i, j), indicating that L-menthol normalized the PP6 downstream pathway.

Notably, topical use of L-menthol did not affect skin homeostasis under normal condition (Supplementary Fig. 3a–g). To further investigate the role of L-menthol in treating psoriasis, we first induced the skin inflammation in mice with IMQ for 5 days and then treated these mice with topical use of L-menthol (Supplementary Fig. 4a). Consistent with previous results, L-menthol relieved skin inflammation and upregulated Pp6 in the epidermis (Supplementary Fig. 4b–g).

Interleukin-17 (IL-17) was reported to downregulate PP6 in psoriatic keratinocytes[19]. Interestingly, L-menthol rescued the expression of PP6 in IL-17-stimulated NHEK cells (Supplementary Fig. 5a). Besides, L-menthol decreased the epidermal thickness and upregulated PP6 in psoriasiform 3D human skin equivalents[21] induced by TNF-α and IL-22 (Supplementary Fig. 5b–e). Taken together, L-menthol upregulates PP6 in inflamed keratinocytes and relieves psoriasis-like skin inflammation.

To investigate whether Pp6 was indispensable for L-menthol's therapeutic effects on psoriatic skin lesions, we crossed mice with loxP-flanked *Pp6* alleles (*Pp6*^fl/fl) with Keratin 5-Cre (K5) mice to delete Pp6 in keratinocytes (designated *K5. Pp6*^fl/fl, Supplementary Fig. 6a–d), then treated mice with IMQ after the topical application of L-menthol or vehicle (EtOH) on the ears for 7 consecutive days (Supplementary Fig. 6e). The topical application of L-menthol improved the psoriasis-like skin phenotype (Fig. 2a) in *Pp6*^fl/fl mice. The protein level of Pp6 in the epidermis derived from *Pp6*^fl/fl mice was dramatically upregulated by L-menthol (Fig. 2b). Histological examination of the skin lesions (Fig. 2c) showed that the epidermal thickness (acanthosis) (Fig. 2d), dermal cellular infiltrates (Fig. 2e) and epidermal Ki67+ cell counts (Fig. 2f, g) were significantly reduced in *Pp6*^fl/fl mice treated with L-menthol compared with vehicle-treated controls. In contrast to *Pp6*^fl/fl mice, L-menthol failed to reduce skin inflammation, acanthosis, dermal cellular infiltrates, epidermal Ki67+ cell counts and inflammatory cytokines in skin lesions derived from *K5. Pp6*^fl/fl mice (Fig. 2c–g and Supplementary Fig. 6f). Taken together, these data demonstrate that L-menthol relieves skin inflammation via upregulating PP6 in keratinocytes.

### L-menthol targets and upregulates HES1 in keratinocytes

To elucidate the PP6-upregulation mechanism of L-menthol, we took advantage of an unbiased target identification approach, DARTS[22]. HaCaT cells were used as the protein source for DARTS assay (Supplementary Fig. 7a). Mass spectrometry identified HES1 among the most abundant and enriched proteins present in the L-menthol treated sample (Supplementary Fig. 7b and Supplementary Data 1). The interaction between HES1 and L-menthol was verified in HaCaT cells (Fig. 3a), primary mouse keratinocytes (Fig. 3b), IMQ-induced mouse epidermis and purified HES1 protein (Supplementary Fig. 7c, d). Furthermore, by performing a microscale thermophoresis (MST) assay, we found that the equilibrium dissociation constant ($K_d$) of the HES1-L-menthol interaction was 16.8 nM (Fig. 3c). Together, these data suggest that L-menthol directly interacts with HES1 in keratinocytes.

Next, we questioned whether HES1 was a potential drug target for psoriasis. We performed single-cell RNA sequencing (scRNA-seq) on the epidermis derived from healthy donors and psoriasis patients (Fig. 3d). HES1 was highly expressed in Keratin 1 (K1)-positively differentiated keratinocytes (Fig. 3e). Compared with that in healthy controls, the number of HES1^hi keratinocytes is significantly decreased in the epidermis of psoriasis patients (Fig. 3e). Furthermore, immunohistochemical analyses showed that HES1 was dramatically decreased in the epidermis of psoriasis patients compared with that of healthy controls (Fig. 3f). Taken together, HES1 is decreased in the psoriatic epidermis and might be a therapeutic target of psoriasis.

To investigate the effect of L-menthol on HES1, we detected the expression of HES1 in the epidermis derived from untreated mice, IMQ-induced mice and IMQ-induced mice treated with L-menthol. HES1 was decreased in IMQ-induced mouse epidermis compared to that in untreated mice, and L-menthol treatment rescued the expression of HES1 in IMQ-induced epidermis (Fig. 3g). Furthermore, we detected the expression of HES1 in HaCaT cells treated with L-menthol in the presence of R848 and found that L-menthol increased the protein level of HES1 in keratinocytes without affecting its mRNA level (Fig. 3h and Supplementary Fig. 7e). Cycloheximide (CHX) was reported to inhibit the eukaryotic translation elongation[23]. The CHX tracking experiment showed that L-menthol prolonged the half-life of HES1 in the presence of CHX, indicating that L-menthol inhibited the degradation of HES1 (Supplementary Fig. 7f). The ubiquitin-proteasome system was reported to mediate the degradation of HES1 protein[24], and thus we hypothesized that L-menthol upregulated HES1 in keratinocytes by inhibiting its proteasomal degradation. We treated R848-stimulated HaCaT cells with MG132, a proteasome inhibitor. L-Menthol treatment did not cause an additional increase of HES1 in the presence of MG132 (Fig. 3i), indicating that L-menthol increases HES1 by inhibiting its proteasomal degradation in keratinocytes. Furthermore, L-menthol led to a decrease in the amount of ubiquitinated HES1 in

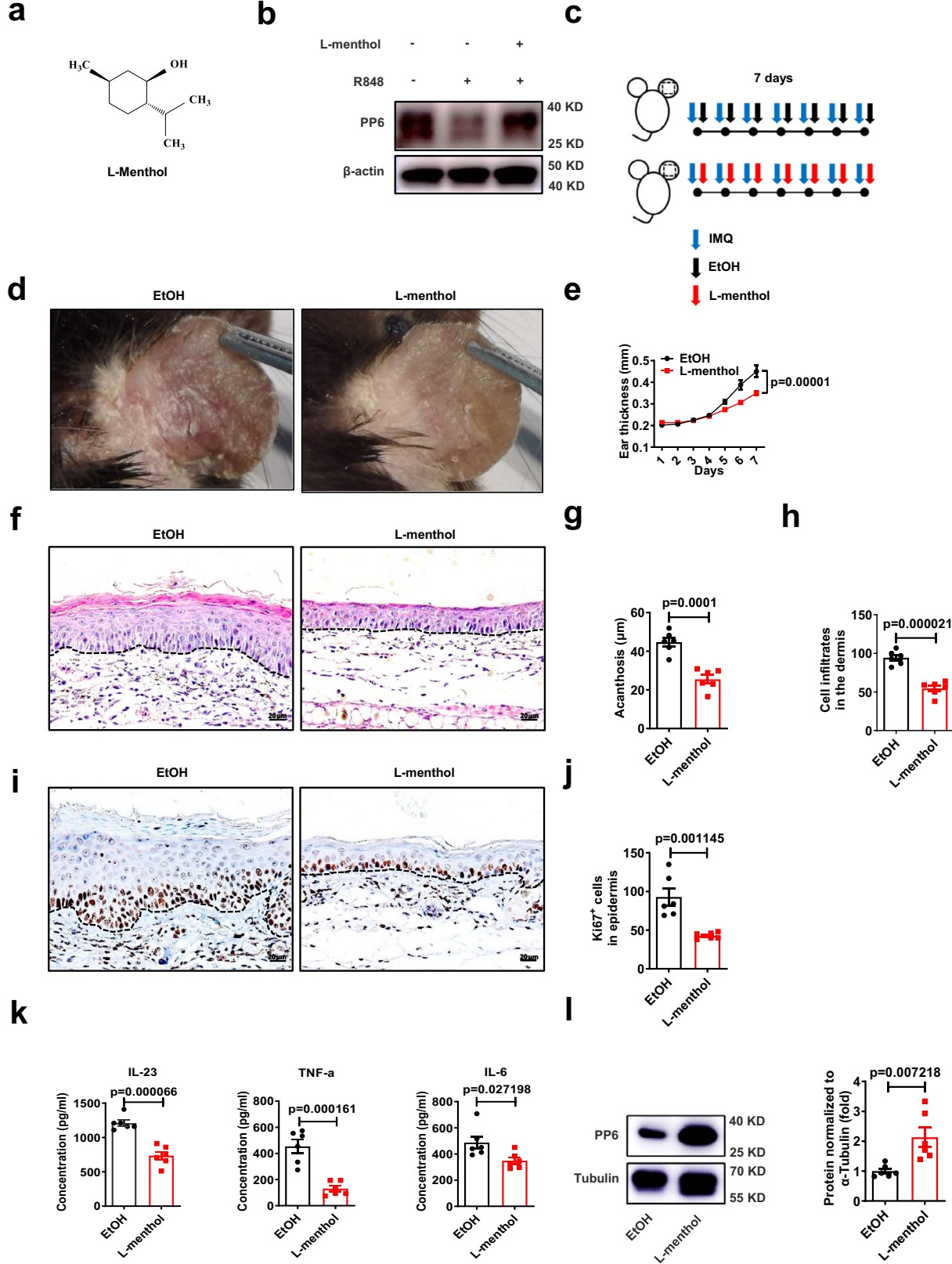

inflamed keratinocytes (Supplementary Fig. 7g). Together, L-menthol targets and upregulates HES1 in keratinocytes.

## HES1 governs the PP6-upregulating function of L-menthol

To further investigate the function of Hes1 in keratinocytes, we crossed mice with loxP-flanked *Hes1* alleles (*Hes1*^fl/fl) with Keratin 5-Cre (*K5*) mice to delete Hes1 (designated *K5. Hes1*^fl/fl). Hes1-deletion in keratinocytes didn't affect skin homeostasis (Supplementary Fig. 8a–f). However, epidermal-specific deletion of Hes1 exacerbated the skin inflammation in the IMQ-induced mouse model of psoriasis (Supplementary Fig. 8a). Histological examination of the skin lesions showed that the epidermal thickness (acanthosis) and dermal cellular infiltrates in skin lesions were

**Fig. 1 | L-menthol upregulates PP6 in inflamed keratinocytes and relieves IMQ-induced skin inflammation. a** Structural formula of L-menthol. **b** Immunoblotting of PP6 in HaCaT cells treated with or without 10 μM L-menthol for 12 h in the presence or absence of 1 μg/ml R848. **c** Schematic of IMQ-induced psoriasis mouse model and 50 μg/day L-menthol treatment. **d** Representative photographs of the ears of ethanol (EtOH) or 50 μg/day L-menthol treated IMQ-induced mice ($n = 6$). **e** Ear thickness changes of IMQ-induced mice treated with EtOH or L-menthol ($n = 6$). **f** Representative hematoxylin and eosin (H&E) staining of the ears treated as in (**d**). Scale bar, 20 μm. **g, h** Acanthosis (**g**) and dermal cellular infiltrates (**h**) of ears treated as in (**d**) ($n = 6$). **i** Representative immunohistochemical staining of Ki67 in ears treated as in (**d**) ($n = 6$). Scale bar, 20 μm. **j** Quantitation of Ki67+ epidermal cells in ears treated as in (**d**). **k** Enzyme-linked immunosorbent assay (ELISA) for IL-23, TNF-α, and IL-6 in skin lesions derived from IMQ-induced mice with the treatment of EtOH or 50 μg/day L-menthol ($n = 6$). **l** Immunoblotting (left panel) and quantitation (right panel) of Pp6 expression in epidermis derived from EtOH or 50 μg/day L-menthol-treated mice after induction of skin phenotype by IMQ ($n = 6$). The data in (**b, d–l**) are representative of three independent experiments. Statistical analyses were performed by two-way ANOVA (**e**) and two-tailed Student's *t*-test (**b, d, f–l**). All data were presented as mean values ± SEM. Specific *p* values are indicated in the figure. Source data are provided as a Source Data file.

significantly increased in IMQ-treated *K5. Hes1*$^{fl/fl}$ mice compared with IMQ-treated *Hes1*$^{fl/fl}$ mice. (Supplementary Fig. 8b–d). Meanwhile, epidermal Ki67+ cell counts in skin lesions derived from IMQ-treated *K5. Hes1*$^{fl/fl}$ mice were significantly augmented compared with IMQ-treated *Hes1*$^{fl/fl}$ mice (Supplementary Fig. 8e, f). In contrast to IMQ-treated *Hes1*$^{fl/fl}$ mice, lower protein but not the mRNA level of Pp6 was found in the epidermis of IMQ-induced *K5. Hes1*$^{fl/fl}$ mice (Supplementary Fig. 8g, h). Furthermore, silencing of HES1 expression in HaCaT cells led to the decrease of protein but not the mRNA level of PP6 (Supplementary Fig. 8i–k). Meanwhile, HES1 downregulation led to the hyper-proliferation of NHEK cells (Supplementary Fig. 8l, m). Taken together, Hes1-deletion in keratinocytes decreases Pp6 and exacerbates IMQ-induced psoriasis-like skin inflammation.

To further determine whether L-menthol upregulated Pp6 through Hes1 in keratinocytes, we treated *Hes1*$^{fl/fl}$ mice and *K5. Hes1*$^{fl/fl}$ mice with IMQ after topical application of L-menthol or vehicle (EtOH) on the ears for 7 consecutive days (Fig. 4a). L-Menthol significantly upregulated Pp6 in the epidermis derived from *Hes1*$^{fl/fl}$ mice (Fig. 4b) and improved the psoriasis-like skin phenotype in *Hes1*$^{fl/fl}$ mice (Fig. 4c). Histological examination of the skin lesions (Fig. 4d) showed that the epidermal thickness (acanthosis) (Fig. 4e) and dermal cellular infiltrates (Fig. 4f) in skin lesions were significantly reduced in *Hes1*$^{fl/fl}$ mice treated with L-menthol compared with vehicle-treated controls. Meanwhile, the topical application of L-menthol dramatically decreased epidermal Ki67+ cell counts (Fig. 4g, h) in *Hes1*$^{fl/fl}$ mice. In contrast, L-menthol failed to increase Pp6 (Fig. 4b) in the epidermis of *K5. Hes1*$^{fl/fl}$ mice nor improve skin inflammation in *K5. Hes1*$^{fl/fl}$ mice (Fig. 4c). L-Menthol also failed to reduce acanthosis (Fig. 4e), dermal cellular infiltrates (Fig. 4f) and epidermal Ki67+ cell counts (Fig. 4g, h) in skin lesions of *K5. Hes1*$^{fl/fl}$ mice. Taken together, HES1 is indispensable for L-menthol to upregulate PP6 in keratinocytes.

## HES1 transcriptionally activates IGBP1 expression

Because HES1 maintained the PP6 protein level without affecting its mRNA level, we asked if HES1 directly interacted with PP6 in keratinocytes. We treated PP6-overexpressing HaCaT cells with or without L-menthol in the presence of R848 and investigated PP6-interactive proteins. Through mass spectrometry on immunoprecipitates pulled by PP6 antibodies and GeneMANIA analysis[25], we identified IGBP1, SET, and AURKA as PP6-interactive proteins, while HES1 was not found to interact with PP6 (Fig. 5a, Supplementary Fig. 9a, and Supplementary Data 2). Furthermore, L-menthol significantly increased IGBP1 but not SET or AURKA in keratinocytes (Fig. 5b and Supplementary Fig. 9b). To assess whether L-menthol upregulated Igbp1 through Hes1 in keratinocytes, we treated IMQ-induced *Hes1*$^{fl/fl}$ mice and *K5. Hes1*$^{fl/fl}$ mice with the topical application of L-menthol or vehicle (EtOH) on the ears for 7 consecutive days. L-Menthol increased Hes1 and Igbp1 in *Hes1*$^{fl/fl}$ mouse epidermis, but failed to increase both proteins in *K5. Hes1*$^{fl/fl}$ mouse epidermis (Fig. 5c), indicating that L-menthol upregulated Igbp1 through Hes1 in keratinocytes. Furthermore, the mRNA level of Igbp1 was lower in the epidermis of *K5. Hes1*$^{fl/fl}$ mice compared with *Hes1*$^{fl/fl}$ mice after IMQ induction (Fig. 5d), indicating that Hes1 might transcriptionally increase Igbp1. We predicted a high-score HES1-binding site in the promoter region of the human *IGBP1* gene with JASPAR[26], and

generated reporter constructs that contained WT or HES1-binding-site-mutated *IGBP1* promoter (Fig. 5e). Silencing of HES1 in HaCaT cells harboring WT *IGBP1* promoter resulted in decreased luciferase activity, and HES1-binding-site-mutation presented parallel luciferase activity. As predicted, silencing of HES1 in HaCaT cells did not additionally reduce the luciferase activity of the HES1-binding-site-mutated reporter construct (Fig. 5f), indicating that HES1 transcriptionally regulated the expression of IGBP1 through binding to its promoter region. Furthermore, HES1-overexpression in HaCaT cells significantly promoted the transcription of Igbp1 but not SET or AURKA (Fig. 5g). Next, we explored the mechanism of how HES1 promoted the transcription of IGBP1. It was previously shown that CaMK2d switched HES1 from a transcriptional repressor to a transcriptional activator by phosphorylation of S126[27,28]. We mutated the serine residue of the CaMK2d phosphorylation site in HES1 to inhibit Hes1-CaMK2d cooperation. Compared with the wild-type HES1, the kinase domain mutated (S126A) form of HES1 did not alter the mRNA level of IGBP1, indicating that the phosphorylation of S126 in HES1 is dispensable for the activation of IGBP1 in keratinocytes (Revised Supplementary Fig. 9c–d). Together, HES1 transcriptionally activates the expression of IGBP1 in keratinocytes, which is found to be an interacting protein of PP6 in keratinocytes.

HES1-overexpression in HaCaT cells led to the dramatic upregulation of the protein level of IGBP1 and PP6 in the presence of R848 (Fig. 5h). To further investigate whether HES1 upregulation suppressed psoriasis pathology, we injected a Hes1-expressing adeno-associated virus serotype 9 (AAV-Hes1) intracutaneously to increase Hes1 in the epidermis of IMQ-induced mice (Supplementary Fig. 10a, b). The epidermal thickness, dermal cellular infiltrates, and epidermal Ki67+ cell counts in skin lesions were significantly decreased in IMQ-induced mice given AAV-Hes1 compared with mice administered AAV-Control (Supplementary Fig. 10c–g). Meanwhile, the protein levels of Hes1, Igbp1 and Pp6 were increased in the epidermis of IMQ-induced mice given AAV-Hes1 (Supplementary Fig. 10h). Thus, HES1 upregulation suppressed the psoriasis-like skin inflammation in mice.

## HES1 increases PP6 through IGBP1 in keratinocytes

Immunohistochemical analyses showed that IGBP1 was dramatically decreased in the epidermis derived from psoriasis patients compared with healthy controls (Supplementary Fig. 11a). We also found that IGBP1 was significantly downregulated in the epidermis of IMQ-induced mice, while topical application of L-menthol rescued the expression of IGBP1 (Supplementary Fig. 11b). To determine whether HES1 regulated PP6 through IGBP1 in keratinocytes, we overexpressed IGBP1 in HaCaT cells transfected with HES1 siRNA (si-HES1) and found that forced IGBP1 expression rescued PP6 which was downregulated upon silencing of HES1 (Fig. 6a, b). Thus, IGBP1 is essential for HES1 to maintain PP6 protein levels.

We next investigated the effect of IGBP1 on PP6 in keratinocytes. A previous study found that IGBP1 deletion leads to the loss of PP6 in the liver[29]. Consistently, we found that silencing of IGBP1 resulted in significantly decreased PP6 in keratinocytes (Fig. 6c). IGBP1 was also reported to promote PP2A-C expression by preventing its ubiquitination[29]. Similarly, we found that downregulation of IGBP1 led to a substantial increase in the amount of ubiquitinated PP6 that could

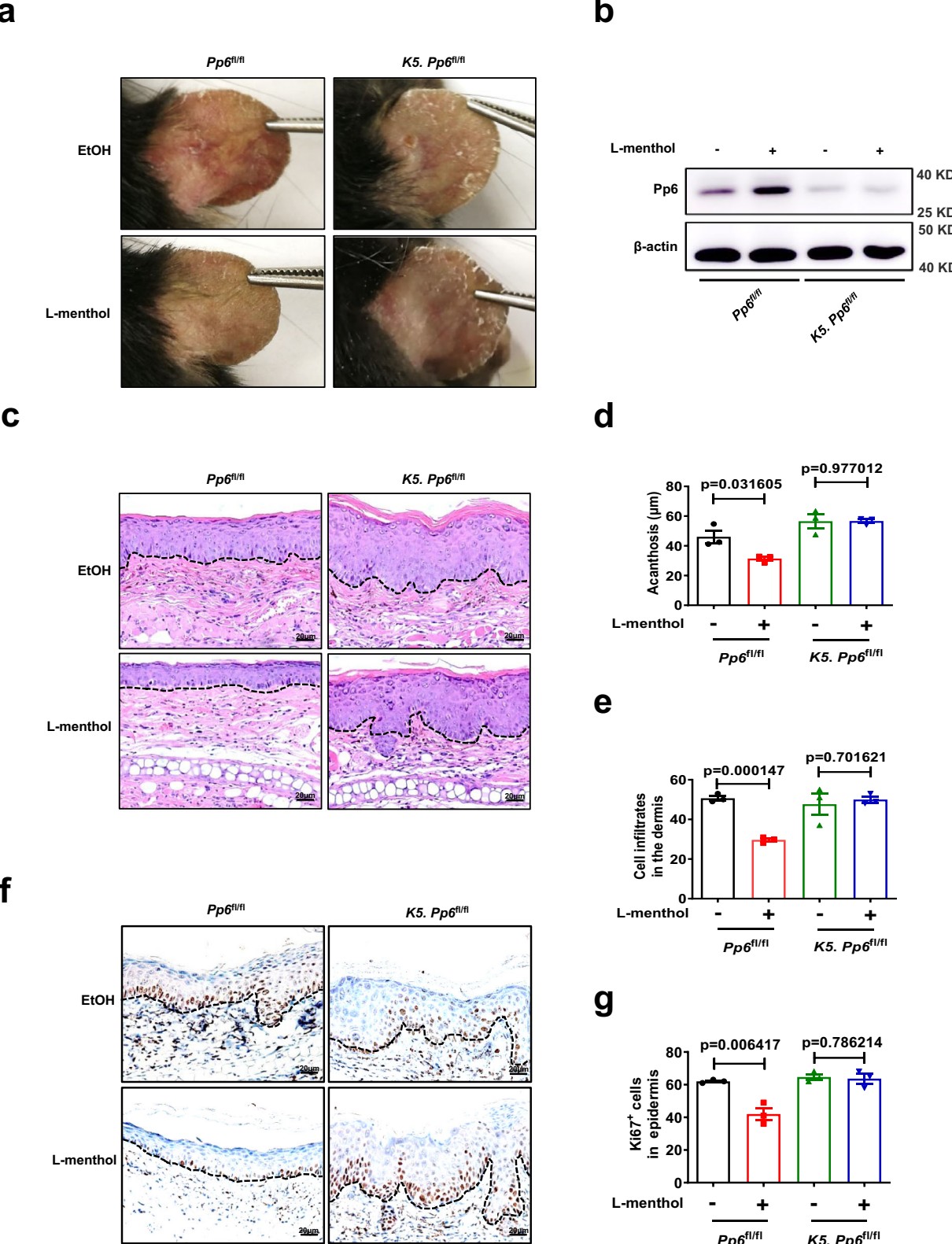

**Fig. 2 | The anti-inflammation effects of L-menthol are mediated by Pp6 in keratinocytes. a** Representative photographs of the ears of IMQ-induced *Pp6*^fl/fl mice and *K5. Pp6*^fl/fl mice treated with EtOH or 50 µg/day L-menthol (*n* = 3). **b** Immunoblotting analysis of Pp6 expression in the epidermis derived from IMQ-induced *Pp6*^fl/fl mice and *K5. Pp6*^fl/fl mice treated with EtOH or 50 µg/day L-menthol. **c** Representative H&E staining of the ears treated as in (**a**) (*n* = 3). Scale bar, 20 µm. **d**, **e** Acanthosis (**d**) and dermal cellular infiltrates (**e**) of ears derived from IMQ-

induced *Pp6*^fl/fl mice and *K5. Pp6*^fl/fl mice treated with EtOH or 50 µg/day L-menthol (*n* = 3). **f** Representative immunohistochemical staining of Ki67 in ears treated as in (**a**) (*n* = 3). Scale bar, 20 µm. **g** Quantitation of Ki67⁺ epidermal cells in ears treated as in (**a**) (*n* = 3). The data in (**a**–**g**) are representative of three independent experiments. Statistical analyses were performed by a two-tailed Student's *t*-test. All data were presented as mean values ± SEM. Specific *p* values are indicated in the figure. Source data are provided as a Source Data file.

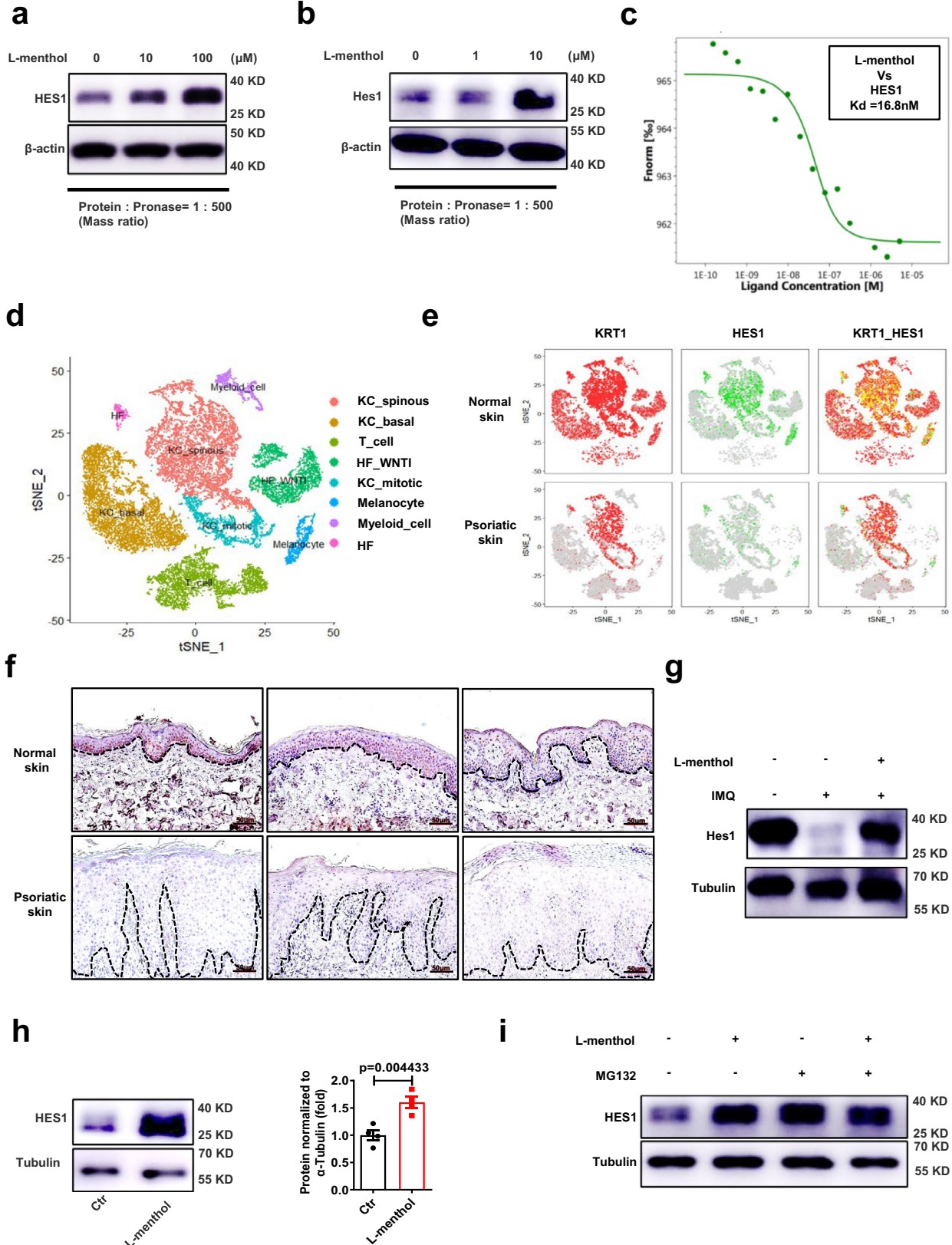

be detected and PP6-immunoprecipitated CHIP (Fig. 6d) which was an E3 ubiquitin ligase reported to regulate the turnover of PP2A subunits[30]. In contrast, overexpression of Igbp1 resulted in dramatically increased PP6 in keratinocytes and a substantial decrease in the amount of ubiquitinated PP6 and PP6-immunoprecipitated CHIP

(Fig. 6e, f). Besides, L-menthol weakened the ubiquitination of PP6 which is consistent with the effect of IGBP1 overexpression in keratinocytes (Supplementary Fig. 11c). Taken together, IGBP1 promotes PP6 expression in keratinocytes and inhibits its ubiquitination probably through disrupting the binding of CHIP.

**Fig. 3 | L-menthol targets and upregulates HES1 in keratinocytes.**
**a** Immunoblotting of HES1 expression in HaCaT cell lysates with or without L-menthol subjected to pronase digestion. **b** Immunoblotting of Hes1 expression in mouse primary keratinocyte lysates with or without L-menthol subjected to pronase digestion. **c** MST analysis of the binding affinity between L-menthol and HES1. **d** Feature plots of indicated genes in merged scRNA-seq datasets from healthy donors ($n = 3$) and patients with psoriasis ($n = 3$). **e** Representative t-SNE plot of scRNA-seq of epidermal cells derived from healthy donors ($n = 3$) and patients with psoriasis ($n = 3$). **f** Immunohistochemical staining of HES1 in skin sections derived from healthy donors ($n = 3$) and patients with psoriasis ($n = 3$). Scale bar, 50 μm.

**g** Immunoblotting of Hes1 expression in epidermis derived from WT mice, IMQ-induced mice, and IMQ-induced mice with 50 μg/day L-menthol treatment. **h** Immunoblotting (left panel) and quantitation (right panel) of HES1 expression in HaCaT cells with or without 10 μM L-menthol treatment in the presence of 1 μg/ml R848 ($n = 4$). **i** Immunoblotting analysis of HES1 expression in HaCaT cells treated with 10 μM L-menthol and 0.5 μM MG132 in the presence of 1 μg/ml R848. The data in (**a–c**, **f–i**) are representative of three independent experiments. Statistical analyses were performed by a two-tailed Student's *t*-test. All data were presented as mean values ± SEM. Specific *p* values are indicated in the figure. Source data are provided as a Source Data file.

## Discussion

PP6 is diminished in the epidermis of psoriasis patients and IMQ-induced mouse model[19], indicating that restoration of PP6 can be a rational strategy for psoriasis treatment. Here, through the phenotypic screening, we identified that L-menthol upregulates the expression of PP6 in inflamed keratinocytes.

L-Menthol, the principal component of peppermint oil, is a natural compound widely used for analgesia and pain relief. L-Menthol activates transient receptor potential (TRP) cation channel subfamily M member 8 (TRPM8)[31] and produces a cooling sensation, which brings welcome relief to psoriasis[32]. L-Menthol also interacts with other TRP channels including TRPA1[33] and TRPV3[34] and may induce central inhibition of nociception through activating GABA(A)-receptors[35]. However, the function of L-menthol in psoriasis remains unclear. We validated that L-menthol ameliorates psoriasis-like skin inflammation through upregulating Pp6 in keratinocytes, indicating that L-menthol harbors the potential to be a drug for psoriasis. Certainly, the modification of L-menthol is worthy of further investigation to enhance its therapeutic effect.

To elucidate the PP6-upregulation mechanism of L-menthol, we adopted a target identification approach, DARTS, and revealed that L-menthol directly targets HES1 in keratinocytes. HES1 belongs to a family of basic helix-loop-helix DNA-binding proteins[36], which attenuates inflammation by regulating transcription elongation[37] and participates in epidermal development[38]. A previous study found that HES1 is upregulated in psoriatic CD34[+] bone marrow cells[39]. Hes1 was also reported to regulate IL-17A[+]γδ[+] T cell expression and IL-17A secretion in mouse psoriasis-like skin inflammation[40]. However, the function of HES1 in psoriatic keratinocytes remains largely unknown. We confirmed that HES1 is significantly decreased in the epidermis of psoriasis patients and IMQ-induced mouse model, which is probably due to the interaction between Wnt5a and Notch1[41]. Hes1 deficiency in keratinocytes significantly down-regulates the expression of Pp6 and exacerbates IMQ-induced skin inflammation, while L-menthol upregulates HES1 by inhibiting its proteasomal degradation. Further studies will be needed to determine the exact HES1-binding mode of L-menthol and the mechanism of varied HES1 expression from different cell types in psoriasis.

Mechanistically, we found that HES1 regulates PP6 in keratinocytes by transcriptionally activating the expression IGBP1. It has been reported that CaMK2d switched HES1 from a transcriptional repressor to a transcriptional activator by phosphorylation of S126[27,28]. We proved that the phosphorylation of S126 in HES1 is dispensable for the activation of IGBP1. Thus, the mechanism of HES1 as a transcriptional activator needs to be further clarified. IGBP1, an essential regulator of PP2A phosphatase activity, was recently reported to form a complex with type 2A protein phosphatase in healthy and hypertrophied myocardium[29,42]. However, how IGBP1 regulates PP6 in keratinocytes remains unknown. We found that IGBP1 overexpression inhibits the ubiquitination of PP6 and decreases a PP6-interacted E3 ligase, CHIP. Further studies will be required to explore how IGBP1 affects the interaction between PP6 and CHIP in keratinocytes.

In summary, we identified a natural compound, L-menthol, for psoriasis therapy by upregulating PP6 in keratinocytes. Using L-menthol as a probe, we uncovered the dysfunctional HES1-IGBP1-PP6 axis that contributed to psoriasis pathology (Fig. 6g). Our preliminary data provide potential therapeutic targets for psoriasis.

## Methods
### Human subjects
Psoriatic skin samples were obtained by punch biopsy from patients who were under local lidocaine anesthesia. Normal adult human skin specimens were taken from healthy donors who were undergoing plastic surgery. All participants provided written informed consent. The study was performed in accordance with the principles of the Declaration of Helsinki and approved by the Research Ethics Boards of Xiangya Hospital of Central South University and Shanghai General Hospital (No. 201311392 and No. 2018KY239).

### Animals
C57BL/6 mice were purchased from Shanghai SLAC Laboratory Animal Co., Ltd. (Shanghai, China). *Pp6*[fl/fl] mice were provided by Dr. Wufan Tao (State Key Laboratory of Genetic Engineering and Institute of Developmental Biology and Molecular Medicine, Fudan University, Shanghai, China)[14]. Keratin 5-Cre transgenic mice were provided by Dr. Xiao Yang (State Key Laboratory of Proteomics, Genetic Laboratory of Development and Disease, Institute of Biotechnology, Beijing, China)[43]. *Hes1*[fl/fl] mice were originally obtained from R. Kageyama (Kyoto University, Kyoto, Japan)[44]. The mice were bred and maintained under specific pathogen-free (SPF) conditions. All mice used in this study were 6–10 weeks old. Mice were used for all of the experiments in accordance with the National Institutes of Health Guide for the Care and Use of Laboratory Animals with the approval (SYXK-2003-0026) of the Scientific Investigation Board of Shanghai Jiao Tong University School of Medicine in Shanghai, China. The catalog number of all diet is cat.1010086 (Jiangsu Xietong Pharmaceutical Bio-engineering Co., Ltd, China). Mice were housed in cages with five mice per cage and kept on in a regular 12-h:12-h light: dark cycle. The temperature was $22 \pm 1\,°C$ and humidity was 40–70%. To ameliorate any suffering that the mice observed throughout these experimental studies, the mice were euthanized by $CO_2$ inhalation.

### Cell culture
To isolate primary murine keratinocytes, the skin from the ears of adult mice was cut into small pieces and then suspended on Dispase II (Sigma-Aldrich cat. D4693, 2 mg/mL) overnight at 4 °C. The following day, the epidermis was peeled and placed into 0.05% trypsin-EDTA (GIBCO cat. 25300062) for 10 min at 37 °C with gentle shaking. The cell suspension was added to a cold trypsin neutralization solution and then centrifuged at $400 \times g$ for 10 min. The cells were cultured in Medium 154CF (GIBCO cat. M154CF500) supplemented with 0.05 mM calcium chloride and human keratinocyte growth supplement (GIBCO cat. S0015). HaCaT cells (Mxbio cat. MXC138) were cultured in DMEM/high glucose (HyClone cat. SH30022.01) containing 10% fetal bovine serum (FBS). The medium was refreshed every 2 days and the cells were sub-cultured according to the cell fusion. Cells in passage 2–6 were used for subsequent experiments. Normal Human Epidermal Keratinocytes (NHEK; Lifeline Cell Technology cat. FC-0007) were cultured in the serum-free basal medium with growth factors (Lifeline

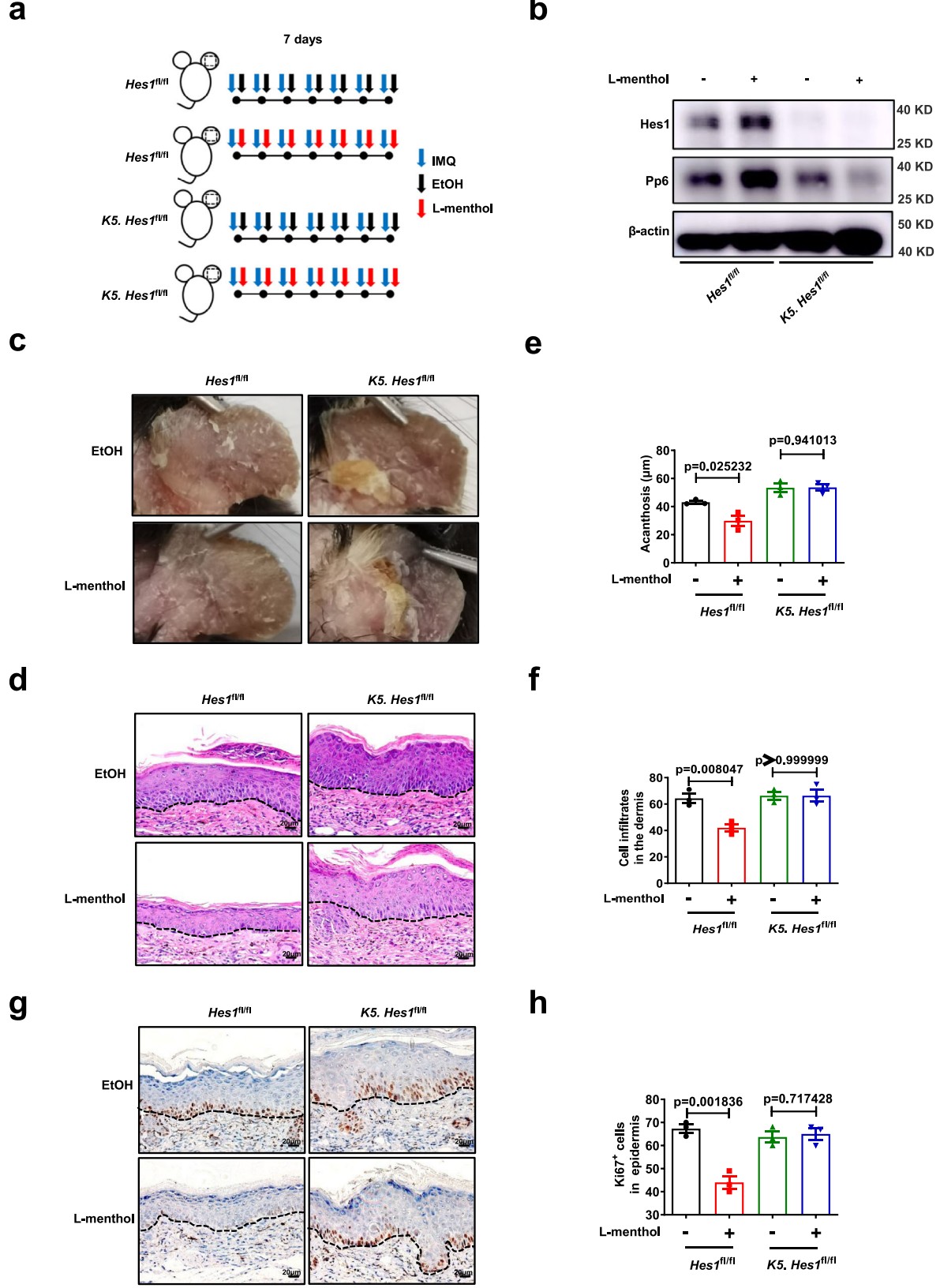

Cell Technology cat. LL-0007). The medium was refreshed every 2 days and the cells were sub-cultured according to the cell fusion.

**Immunohistochemical and histological analysis**
Skin specimens were embedded in paraffin and sectioned at the Histology Core of the Shanghai Institute of Immunology. For immunohistochemical staining, the sections were deparaffinized and washed in phosphate-buffered saline (PBS). Antigen retrieval was performed by heating the sections in 10 mM sodium citrate buffer (pH = 6.0). The sections were washed in PBS after cooling, incubated in 3% hydrogen peroxide for 10 min at room temperature (RT), and then washed again in PBS. The sections were blocked in PBS containing 1%

**Fig. 4 | HES1 governs the PP6-upregulating function of ʟ-menthol in keratinocytes. a** Schematic of IMQ-induced psoriasis mouse model and 50 μg/day ʟ-menthol treatment in *Hes1*^fl/fl mice or *K5. Hes1*^fl/fl mice. **b** Immunoblotting analysis of Hes1 and Pp6 expression in the epidermis derived from IMQ-induced *Hes1*^fl/fl mice and *K5. Hes1*^fl/fl mice treated with EtOH or 50 μg/day ʟ-menthol. **c** Representative photographs of ears from IMQ-induced *Hes1*^fl/fl mice and *K5. Hes1*^fl/fl mice treated with EtOH or 50 μg/day ʟ-menthol (*n* = 3). **d** Representative H&E staining of ears treated as in (**c**) (*n* = 3). Scale bar, 20 μm. **e, f** Acanthosis (**e**) and dermal cellular infiltrates (**f**) of ears treated as in (**c**) (*n* = 3). **g** Representative immunohistochemical staining of Ki67 in ears treated as in (**c**) (*n* = 3). Scale bar, 20 μm. **h** Quantitation of Ki67⁺ epidermal cells in ears treated as in (**c**) (*n* = 3). The data in (**b**–**h**) are representative of three independent experiments. Statistical analyses were performed by a two-tailed Student's *t*-test. All data were presented as mean values ± SEM. Specific *p* values are indicated in the figure. Source data are provided as a Source Data file.

bovine serum albumin (BSA) for 1 h at RT and stained in a blocking buffer containing primary antibodies (anti-PPP6C, Merck Millipore cat. 07-1224, 1:50 dilution; anti-HES1, Cell Signaling Technology cat. 11988 S, 1:100 dilution; anti-IGBP1, Abcam cat. ab170911, 1:100 dilution; anti-mouse Ki67, Servicebio cat. GB121141, 1:500 dilution) overnight at 4 °C. On the following day, the sections were warmed to RT for 1 h, washed three times in PBS, and stained with an HRP-polymer complex for 20 min, which was followed by incubation with a secondary antibody for 20 min. The sections were washed three times in PBS, developed with DAB reagent (Peroxidase Substrate Kit, ZSGB-BIO cat. ZLI-9018), and counterstained with hematoxylin. The sections were washed with tap water, and then subsequent washes of increasing ethanol concentration for dehydration. Once mounted and air-dried, the slices were viewed under a Zeiss Axio Scope.A1 light microscope equipped with an AxioCam MRc digital camera, captured with Axiovision software and analyzed by Adobe Photoshop CS4. Epidermal hyperplasia (acanthosis) was assessed by using an average length of three times of measures from the basal membrane to the cornified layer of the epidermis. Inflammatory cell infiltrates in the dermis were also assessed as a histological feature and were equal to the sum of the cells in three areas (100 pixels × 100 pixels) which were randomly taken in the dermal part of every picture. Ki67⁺ cell numbers were calculated in the epidermal part of every picture.

## Animal models of psoriasis and treatment
For the IMQ-induced mouse model of psoriasis, male C57BL/6 mice (7 weeks of age) were maintained under SPF conditions. The mice were subjected to a daily topical dose of 25 mg per ear for 7 consecutive days. For the therapeutic experiments, 20 μL ethanol (EtOH) containing ʟ-menthol (Sigma-Aldrich cat. 224464, 2.5 μg/μl) was topically applied to mouse ears daily prior to the IMQ treatment for 7 consecutive days. All procedures were approved and supervised by the Shanghai Jiao Tong University School of Medicine Animal Care and Use Committee.

## Immunoprecipitation and immunoblotting
For PP6 immunoprecipitation, nuclear and cytoplasmic extracts from HaCaT cells were prepared with Cell lysis buffer for Western and IP (Beyotime cat. P0013) supplemented with a protease inhibitor cocktail (Bimake cat. B14001). The lysates were incubated with anti-PPP6C (Abcam cat. 131335, 1:50 dilution) or rabbit IgG (Abcam cat. 172730, 1:50 dilution) overnight at 4 °C, followed by binding of protein A/G magnetic beads (Merck Millipore cat. LSKMAGAG02) for 4 h at 4 °C. The beads were rinsed three times with a wash buffer and then eluted with an SDS loading buffer. The immune precipitates were then subjected to coomassie blue staining (Beyotime cat. P0017). For immunoblotting, mouse skins or cultured cells were lysed in Cell lysis buffer for Western and IP (Beyotime cat. P0013) containing a protease and phosphatase inhibitor cocktail (Thermo Fisher Scientific cat. 78440). Anti-PPP6C (Merck Millipore cat. 07-1224, 1:1000 dilution), anti-β-actin (Proteintech cat. 60008-1-Ig, 1:4000 dilution), anti-HES1 (Cell Signaling Technology cat. 11988 S, 1:1000 dilution), anti-IGBP1 (Abcam cat. ab170911, 1:1000 dilution), anti-alpha Tubulin (Proteintech cat. 66031-1-Ig, 1:2000 dilution), anti-CHIP (Abcam cat. ab134064, 1:1000 dilution), anti-HA-Tag (Proteintech cat. 66006-2-Ig, 1:10000 dilution), anti-SET (Abcam cat. 181990, 1:1000 dilution), anti-AURKA (Abcam cat. 247643, 1:1000 dilution), anti-Ubiquitin (Cell Signaling Technology

cat.3936 S, 1:1000 dilution), anti-TLR7 (Novus cat. NBP2-27332, 1:100 dilution), anti-C/EBP-β (Abcam cat. 32358, 1:1000 dilution), anti-p-C/EBP-β (Thr188) (Cell Signaling Technology cat. 3084, 1:1000 dilution), HRP-labeled goat anti-mouse IgG (H + L) (Beyotime cat. A0216, 1:1000 dilution), and HRP-labeled goat anti-rabbit IgG (H + L) (Beyotime cat. A0208, 1:1000 dilution) were used as antibodies. The signal was detected with ECL Western Blotting Substrate (Thermo Fisher Scientific cat. 34095) and an Amersham Imager 600 (GE Healthcare). The images were cropped for presentation.

## Screening for PP6-upregulating compounds
HaCaT cells were cultured in a 24-well plate and treated with DMSO or 10 μM natural compounds from the library (Tsbiochem, cat. L-6000) for 12 h in the presence of 1 μg/ml R848. HaCaT cells were lysed with cell lysis buffer for Western and IP (Beyotime cat. P0013) supplemented with a protease inhibitor cocktail (Bimake cat. B14001). The expression of PP6 was detected by immunoblotting.

## qPCR
Skin biopsy specimens were snap-frozen in liquid nitrogen and pulverized. HaCaT cells were rinsed with PBS. Total RNA was isolated using TRIzol Reagent (Invitrogen cat. 15596026) and quantified with a NanoDrop spectrophotometer. For qPCR detection, total RNA was reverse-transcribed into cDNA using HiScript II Q RT SuperMix for qPCR (+gDNA wiper) (Vazyme Cat.R22301). qPCR was conducted with the Hieff qPCR SYBR Green Master Mix (Yeasen Biotech cat. 11202ES03) in a ViiA 7 Real-Time PCR system (Applied Biosystems). The relative expression of target genes was confirmed using the quantity of target gene/quantity of GAPDH. Sequences of primers are provided in the Supplementary Table.

## Constructs
pcDNA3.1 was used to generate the expression constructs of IGBP1, HES1 with a flag-tag, the kinase domain mutated (S126A) form of HES1, ubiquitin with an HA-tag, and PP6 with a C-Myc-Tag for protein interaction studies. pET-28a (+) was used for the expression construct of HES1 with a His-Tag for MST. pSUMO3 was used for the expression construct of HES1 for DARTS.

## ELISA
To detect murine IL-6, TNF-α, and IL-23 in skin lesions, a Mouse IL-6 ELISA kit (NeoBioscience cat. EMC004.96), a Mouse TNF-α ELISA kit (NeoBioscience cat. EMC102a.96), and a mouse IL-23 ELISA kit (BioLegend cat. 433704) were used.

## Microscale thermophoresis (MST) assay
To test the binding affinity of HES1 and ʟ-menthol, MST assay was conducted by using Monolith NT.115 (NanoTemper Technologies). Proteins were labeled with the Monolith His-Tag Labeling Kit RED-tris-NTA 2nd Generation (NanoTemper Technologies cat.MP-L018) according to the supplied labeling protocol. Labeled proteins were used at a concentration of ~50 nM. ʟ-Menthol was titrated in 1:1 dilution beginning at 5 μM, which contained 5% (v/v) DMSO. Samples were diluted in PBS buffer containing 0.05% Tween supplemented with DMSO at a final concentration of 5% to make sure that all samples contained the same DMSO concentration. For the measurement, the

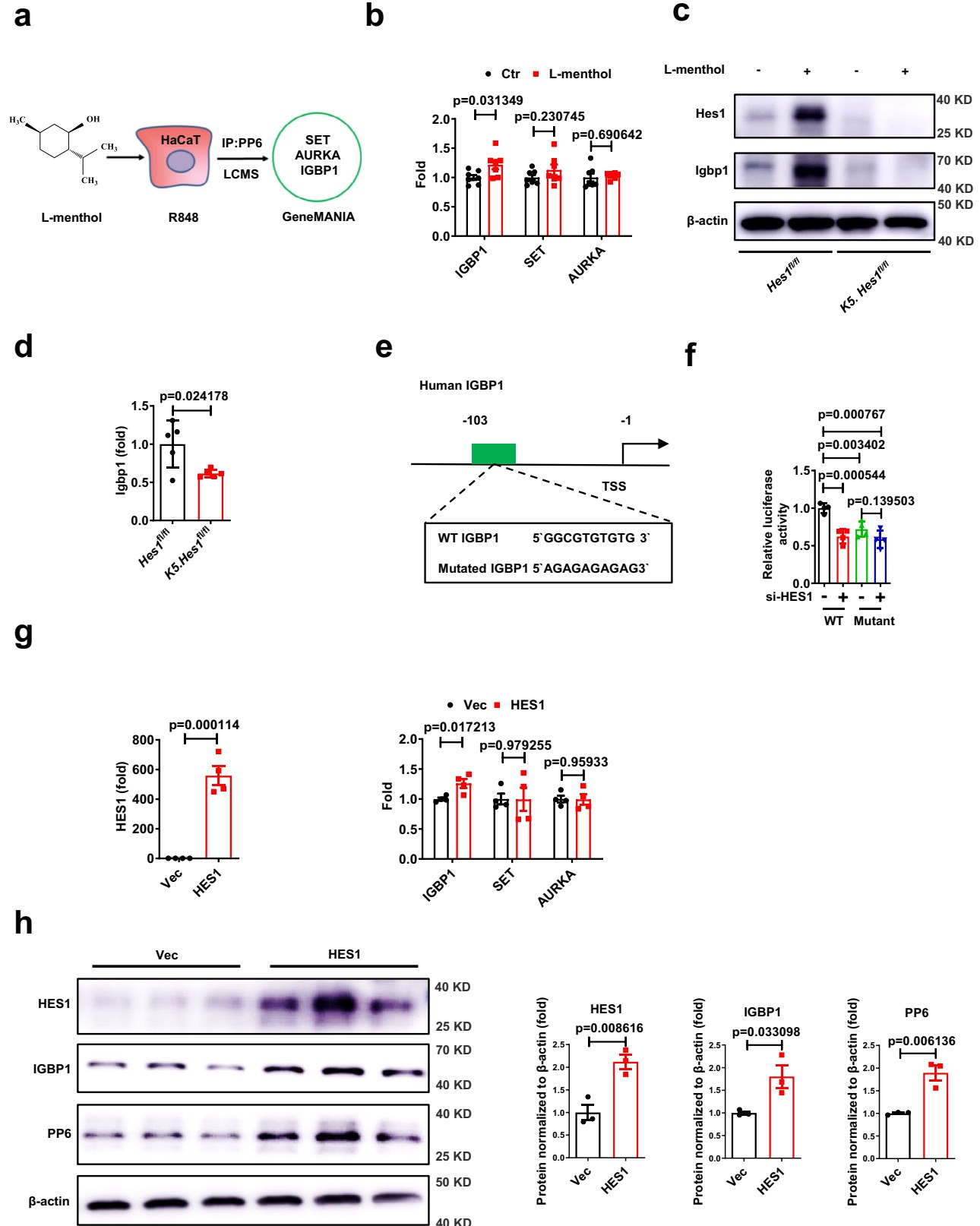

samples were filled into premium-coated capillaries (NanoTemper Technologies cat.MO-K025).

**Drug affinity responsive target stability assay**

A drug affinity responsive target stability (DARTS) assay for identifying the targets of ʟ-menthol was performed according to the protocol by

refs. 22,45. The protein was isolated from HaCaT cells by using Cell lysis buffer (Beyotime cat. P0013) supplemented with a protease inhibitor cocktail (Bimake cat. B14001) and centrifuged at $18,000 \times g$ at 4 °C for 10 min. After centrifugation, $10 \times$ TNC buffer (500 mmol/L Tris-HCl, 500 mmol/L NaCl, and 100 mmol/L CaCl2) was added. The protein concentration was measured using Pierce™ Rapid Gold BCA Protein

**Fig. 5 | HES1 transcriptionally activated the expression of IGBP1. a** PP6-interacted proteins in HaCaT cells treated with 10 μM L-menthol in the presence of 1 μg/ml R848. **b** qPCR detection of indicated genes in HaCaT cells treated with or without 10 μM L-menthol for 12 h in the presence of 1 μg/ml R848. Results are presented as the ratio of indicated genes to the GAPDH relative to that in the control ($n = 7$). **c** Immunoblotting of indicated proteins in the epidermis from IMQ-induced $Hes1^{fl/fl}$ mice and cKO mice treated with or without 50 μg/day L-menthol. **d** qPCR detection of Igbp1 in the epidermis from IMQ-induced $Hes1^{fl/fl}$ mice or cKO mice ($n = 5$). The data were presented as the ratio of Igbp1 to GAPDH relative to that in $Hes1^{fl/fl}$ mice. **e** WT and mutated promoter reporter constructs. HES1-binding site in IGBP1 promoter predicted by JASPR. **f** Luciferase activity in HaCaT cells transfected with or without HES1 siRNA (si-HES1) and the WT or point-mutated promoter reporter plasmid of IGBP1 ($n = 4$). **g** qPCR detection of indicated genes in HaCaT cells transfected with or without a HES1-overexpression plasmid (HES1). The data were presented as the ratio of indicated genes to GAPDH relative to that in control ($n = 4$). **h** Immunoblotting and quantitation of indicated proteins in HaCaT cells transfected with or without a HES1-overexpression plasmid (HES1) in the presence of 1 μg/ml R848. ($n = 3$). The data in (**b**–**h**) are representative of three independent experiments. Statistical analyses were performed by a two-tailed Student's $t$-test. All data were presented as mean values ± SEM. Specific $p$ values are indicated in the figure. Source data are provided as a Source Data file.

Assay Kit (Thermo Fisher Scientific cat. A53225). The protein isolated from HaCaT cells was treated with L-menthol for 30 min at room temperature and then treated with pronase (Roche cat.10165921001) for 20 min at room temperature. After stopping the reaction with 20× protease inhibitor (Bimake cat. B14001), 5× loading buffer (Servicebio cat. G2013) was added and the protein samples were heated at 100 °C for 5 min. Then, the protein samples were resolved by SDS-PAGE. and stained using Pierce™ Silver Stain for Mass Spectrometry (Thermo Fisher Scientific cat.24600). The protected protein bands were further analyzed by matrix-assisted laser desorption/ionization time-of-flight mass spectrometry (MALDI-TOF MS). The interaction between HES1 and L-menthol was verified using HaCaT cells, primary mouse keratinocytes, IMQ-induced epidermis and purified HES1 protein.

## Luciferase reporter assays

For promoter reporter assay, the pGL3-basic vector (Promega, #E1751) was used to clone the promoter of IGBP1. The site-specific mutant was generated by PCR (Koyobo, #SMK-101). Sequences of primers are provided in the Supplementary Table.

HaCaT cells were grown in DMEM (Thermo Scientific, #SH30243.01B) supplemented with heat-inactivated 10% FBS (Gibco, #10437036) in an atmosphere of 5% $CO_2$ at 37 °C. HaCaT cells were seeded in a 24-well plate with a density of $8 \times 10^4$ per well one day before transfection and then each well was transfected with a mixture of 450 ng pGL3 luciferase vector, 50 ng pRL-TK renilla vector and 40 pmol siRNA-Hes1 (GenePharma) using TurboFect (Thermo Scientific cat. R0531). Forty-eight hours post-transfection, cells were lysed and the luciferase activity was measured on a microplate reader (Berthold, TriStar LB941) by using the Dual-Luciferase Reporter Assay System (Promega cat. E1910). The ratio of firefly luciferase to renilla luciferase was calculated for each well.

## RNA interference

Small interfering RNA (siRNA) targeting human IGBP1 or HES1 was synthesized by GenePharma. Sequences of oligonucleotides are provided in the Supplementary Table. The siRNA molecules were transfected using TurboFect (Thermo Scientific cat. R0531)

## scRNA-seq

Fresh skin was placed in saline at 4 °C until further processing. Single-cell suspensions were generated by enzyme digestion and subjected to FACS to exclude doublets, debris, and DAPI-positive dead cells. Sorted cells were centrifuged and resuspended in 0.04% BSA in PBS. Chromium Single Cell 3′ v3 (10x Genomics) library preparation was conducted by the Sequencing Core at the Shanghai Institute of Immunology, according to the manufacturer's instructions. The resulting libraries were sequenced with an Illumina HiSeq 4000 platform. Raw data were processed using Cell Ranger (version 3.0) and further filtered, processed, and analyzed using the Seurat package[46].

## HES1 protein expression and purification

Human HES1 expression plasmid in pET-28a (+) vector was transformed into Escherichia coli BL21 (DE3)-competent cells and expressed as a fusion protein containing a C-terminal 6×His-tag. Cells were grown at 37 °C in 2×YT medium until $OD_{600} = 6.0$–8.0 and expression was induced by the addition of 0.5 mM isopropyl β-ᴅ-1-thiogalactopyranoside (IPTG) overnight at 20 °C. Cells were pelleted and resuspended in cold lysis buffer (10 mM Tris pH 7.4, 0.5 M NaCl, 20 mM imidazole, and 2.5 mM DTT). The cell suspension was lysed with an ultra-high-pressure cell disruptor and insoluble debris was removed by centrifugation at $27,000 \times g$ at 4 °C for 1 h. Proteins were purified by nickel chromatography (Smart-Lifesciences, cat. SA005C15) and eluted with 200 mM imidazole in a lysis buffer. The eluted proteins were further purified by size-exclusion chromatography using a HILOAD 26/60 SUPERDEX 75PG (GE Healthcare). Fresh proteins were used for MST.

Human HES1 expression plasmid in the pSUMO3 vector was transformed into Escherichia coli BL21 (DE3)-competent cells and expressed as a fusion protein containing an N-terminal SUMO-tag. Cells were grown at 37 °C in 2×YT medium until $OD_{600} = 6.0$–8.0 and expression was induced by the addition of 0.5 mM isopropyl β-ᴅ-1-thiogalactopyranoside (IPTG) overnight at 20 °C. Cells were pelleted and resuspended in cold lysis buffer (10 mM Tris pH 7.4, 0.5 M NaCl, 20 mM imidazole, and 2.5 mM DTT). The cell suspension was lysed with an ultra-high-pressure cell disruptor and insoluble debris was removed by centrifugation at $27,000 \times g$ at 4 °C for 1 h. Proteins were purified by nickel chromatography (Smart-Lifesciences, cat.SA005C15) and eluted with 200 mM imidazole in a lysis buffer. SUMO-tag was removed by SENP cleavage overnight by dialysis against 10 mM Tris pH 7.4, 0.5 M NaCl, and 2.5 mM DTT, which was followed by nickel reverse chromatography and dialysis to desalt. Proteins were eluted with lysis buffer and further purified by size-exclusion chromatography using a HILOAD 26/60 SUPERDEX 75PG (GE Healthcare). Fresh proteins were used for DARTS.

## Protein mass spectrometry

Samples from DARTS assay and immunoprecipitates pulled down with anti-PPP6c were analyzed using an Agilent 1290 HPLC that was coupled to an Agilent 6545 Quadrupole Time-of-flight (QToF) mass spectrometer (MS) with a Dual Jet Stream electrospray ionization (ESI) source. The samples were analyzed in positive-ion (ESI⁺) mode. Spectral peaks in the mass range from 50 to 1100 m/z were acquired. The run was performed at a flow rate of 300 μL/min and infused into the ion spray source held at 3.5 kV. Nitrogen was used as the drying gas at 325 °C at a flow rate of 8 L /min. The nebulizer pressure and fragmentor voltage were maintained at 45 psig and 110 V, respectively.

## Quantification and statistical analysis

The data were analyzed using GraphPad Prism 6.02 and are presented as the mean ± SEM. A Student's $t$-test was used to compare two conditions, and an analysis of variance (ANOVA) with Bonferroni or Newman–Keuls correction was used for multiple comparisons. The probability values of <0.05 were considered statistically significant. Exact $p$ value was provided. The error bars depict the SEM.

## Psoriasiform 3D human skin equivalent

3D human skin equivalent and culture medium were purchased from Regenovo (Hangzhou, China). For stimulation, the culture

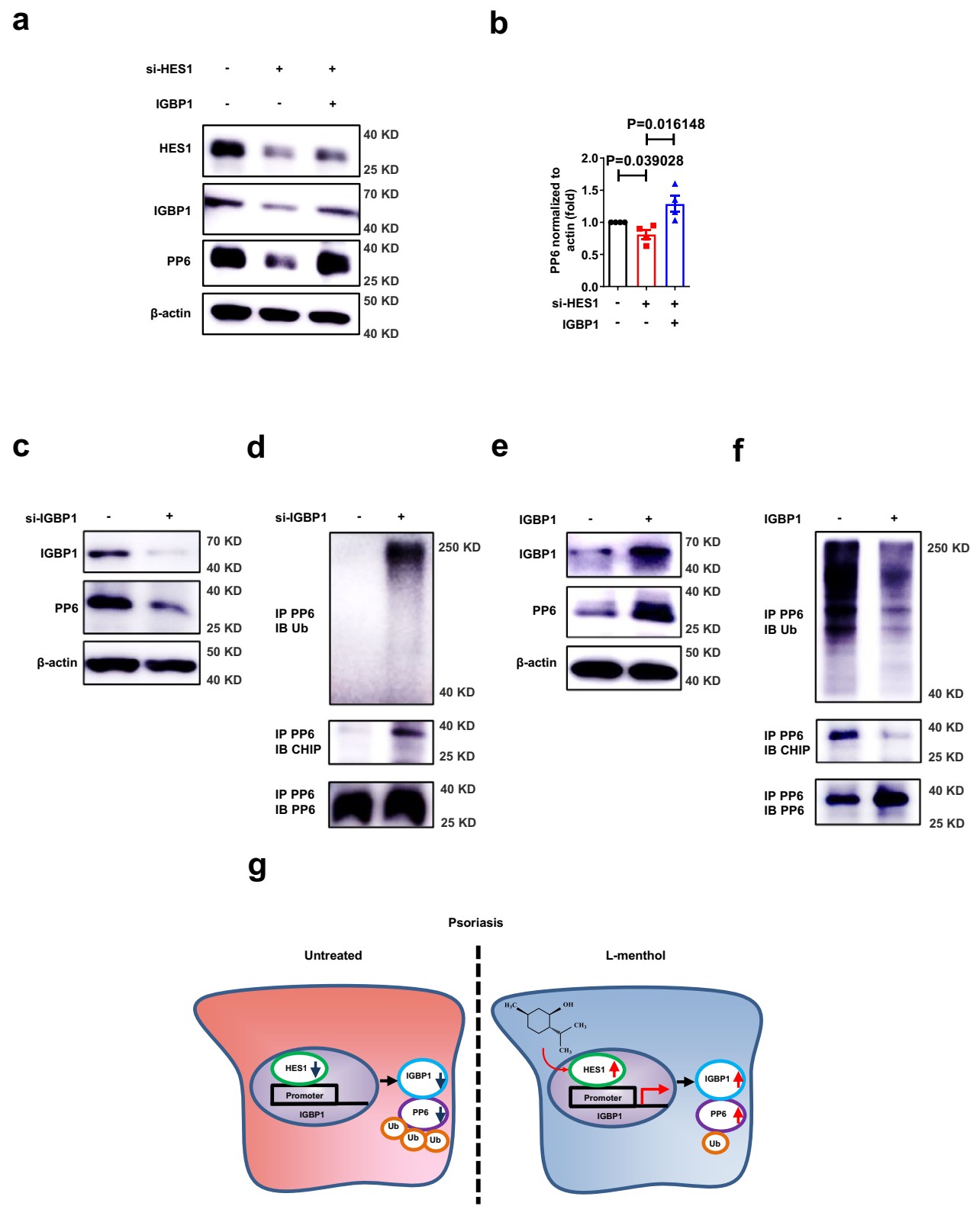

medium was supplemented with a combination of 20 ng/ml IL-22 and 5 ng/ml TNF-α. After 24 h, psoriasiform 3D human skin equivalents were treated with DMSO or 10 μM ʟ-menthol. After 48 h, 3D skin cultures were processed for immunohistochemical analyses and western blot.

### 3D human skin with PP6-knockdown keratinocytes

Adenovirus (AD) vector-mediated downregulation of human PP6 and scrambled control were constructed, amplified, and purified by Hanbio Biotechnology (Shanghai, China). HaCaT cells were infected with the adenovirus at a multiplicity of infection of 100. After 3 days,

**Fig. 6 | IGBP1 promotes PP6 expression in keratinocytes. a** Immunoblotting of HES1, IGBP1, and PP6 expression in HaCaT cells transfected with or without HES1 siRNA (si-HES1) and IGBP1 overexpression plasmid (IGBP1). **b** quantitation of PP6 expression in HaCaT cells transfected with or without HES1 siRNA (si-HES1) and IGBP1 overexpression plasmid (IGBP1) (*n* = 4). **c** Immunoblotting of IGBP1 and PP6 expression in HaCaT cells transfected with scramble siRNA (Ctr) or with IGBP1 siRNA (si-IGBP1). **d** Immunoblotting of PP6-immunoprecipitated proteins in HaCaT cells transfected with an HA-Ubiquitin plasmid and scramble siRNA (Ctr) or with IGBP1 siRNA (si-IGBP1). **e** Immunoblotting of IGBP1 and PP6 expression in HaCaT cells transfected with an empty vector (Vec) or an IGBP1 overexpression plasmid (IGBP1). **f** Immunoblotting of PP6-immunoprecipitated proteins in HaCaT cells transfected with an HA-Ubiquitin plasmid and an empty vector (Vec) or an IGBP1 overexpression plasmid (IGBP1). **g** Cartoon sketch of the molecular mechanism of L-menthol in psoriasis therapy. The data in (**a**–**f**) are representative of three independent experiments. Statistical analyses were performed by a two-tailed Student's *t*-test. All data were presented as mean values ± SEM. Specific *p* values are indicated in the figure. Source data are provided as a Source Data file.

HaCaT cells were subjected to immunofluorescent staining, western blot, or 3D skin culture. About $2 \times 10^5$ HaCaT cells were seeded on the transwell (CORNING cat. 3493) in DMEM/high glucose (HyClone cat. SH30022.01) containing 10% fetal bovine serum (FBS) in a 24 wells format. Culture medium was refreshed every day. After 3 days, cultures were switched to CnT-PR-3D medium (CELLNTEC cat. Cnt-PR-3D) for 24 h and then cultured at the air-liquid interface for 6 days. Culture medium was refreshed every other day. After 6 days of air-exposure, 3D skin cultures were processed for immunohistochemical analyses.

### The MTT cell proliferation assay with HES1-knockdown keratinocytes

Adenovirus (AD) vector-mediated downregulation of human HES1 and scrambled control were constructed, amplified, and purified by Hanbio Biotechnology (Shanghai, China). NHEK cells were infected with the adenovirus at a multiplicity of infection of 100. After 2 days, NHEK cells were subjected to western blot, or MTT cell proliferation assay (Beyotime cat. C0009S). About $1.5 \times 10^4$ NHEK cells were seeded in the 96-well plate. The next day, 10 μl MTT solutions from the Stock (5 mg/ml) was added and cells were incubated in $CO_2$ incubator in the dark for 4 h. 100 μl Formazan dissolved solution was added and cells were incubated in $CO_2$ incubator in the dark for 3 h. The absorbance was read at 570 nm on a Multiskan GO (Thermo Fisher Scientific).

### Local delivery of adeno-associated virus (AAV) in ear skin

Adeno-associated virus (AAV) vector-mediated overexpression of mouse Hes1 and scrambled control (GFP) were constructed, amplified, and purified by Hanbio Biotechnology (Shanghai, China). A total of 20 μl of $1 \times 10^{12}$ vg/μl of each AAV diluted in PBS was injected into the ears of mice intracutaneously. After 18 days of the injection, mice were treated with IMQ for 7 consecutive days, then mice were sacrificed and tissues were dissected for further analysis.

### Cycloheximide (CHX) tracking experiment

HaCat cells were treated with 1 μg/ml R848 and DMSO or 10 μM L-menthol along with 25 μg/ml cycloheximide (MedChem Express cat. HY12320) for the indicated time before being collected with cell lysis buffer (Beyotime cat. P0013) supplemented with a protease inhibitor cocktail (Bimake cat. B14001). Cell lysates were subjected to immunoblot analysis.

### Reporting summary

Further information on research design is available in the Nature Portfolio Reporting Summary linked to this article.

### Data availability

The single-cell RNA-sequencing data in this study have been deposited in the Genome Sequence Archive (GSA) with accession number HRA003418. The mass spectrometry proteomics data have been deposited to the ProteomeXchange Consortium via the PRIDE partner repository with the dataset identifier PXD037335 and PXD037336. The other data supporting the findings of this study are available within the paper or the supplementary materials. Source data are provided with this paper.

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

## Acknowledgements

We would like to thank Dr. XH (Institute for Immunology and School of Medicine, Tsinghua University, Beijing, China) and Prof. Hiroshi Kageyama (Kyoto University, Kyoto, Japan) for providing Hes1fl/fl mice. Our proteomics and mass spectrometry work were performed at the Proteomics Platform of Core Facility of Basic Medical Sciences, Shanghai Jiao Tong University School of Medicine (SJTU-SM). We also thank the staff members from the National Facility for Protein Science in Shanghai (NFPS), Zhangjiang Lab, China for providing technical support in the MST assay. This work was supported by grants from the National Natural Science Foundation of China (No. 82050009, 81930088, 82073428, 82173417, 82103719, and 82101909), National Key Research and Development Program of China Stem Cell and Translational Research (2020YFA0112900), Clinical Research Plan of Shanghai Shenkang Hospital Development Center (SHDC2020CR3061B), SJTU Trans-med Awards Research (20210102), Integrated innovation fund of Shanghai Jiaotong University (2021JCPT04), Experimental Animal Research Project of "Scientific and Technological Innovation Action Plan" (22140903100), and Innovative Research Team of High-Level Local Universities in Shanghai.

## Author contributions

Z.W. and H.W. designed the research, analyzed the data, and wrote the paper. Z.W. conducted most of the experiments; Y.S., F.L., J.B., X.C., Q.Y., L.S., S.T., H.Z., Y.W., L.F., Z.X., and H.W. helped with the experimental details. X.H. helped to breed the *Hes1*fl/fl mice. H.W. supervised the study.

## Competing interests

The authors declare no competing interests.
