## [Peer Review File · Nature Communications]

Targeting HES1 by L-menthol Restores PP6 in Keratinocytes for Psoriasis TherapyREVIEWER COMMENTS

Reviewer #1 (Remarks to the Author):

In this study, the authors identified that the natural compound L-menthol improves psoriatic skin inflammation by up-regulating Pp6 in keratinocytes, and revealed that L-menthol targets Hes1 and up-regulates the co-binding protein IGBP1 of Pp6, which inhibits the degradation of Pp6. This is a novel study that validates L-menthol as a potential drug for psoriasis. The whole experimental design is clear. However, some weaknesses in the data and its representation should be addressed.

Specific comments:

1.Regarding the design of animal experiment, the IMQ-induced psoriasis mice were treated with L-menthol at the same time of modeling for 7 days.Whether there is any influence on the modeling effect when using similar topical administration methods? In this case, is the role of drug intervention to protect or treat?

2.In Supplementary Figure 1C, 20 μ M L-menthol seems to be less effective in stimulating PP6 than 10 μ M. Why? Figure S1 E and F show that different concentrations of L-menthol have completely different effect on acanthosis and dermal cell infiltration. Consequently, the following experiments showing that L-menthol relieves skin inflammation should specify the specific concentration of L-menthol.

3.Figures 3A, 3B and S3C verify the interaction of HES1 and L-menthol in HaCaT cells, primary mouse keratinocytes and purified HES1 protein. It should also be better validated in the keratinocytes of mice with psoriasis.In addition to regulating the proteasome degradation of HES1(figure 3I), does L-menthol affect the protein translation process of HES1? CHX tracking experiment should be designed here.Does L-menthol affect the protein level of HES1 ubiquitination?

4.Regarding the regulation study between HES1 and PP6, Figure 4 demonstrate the up-regulation effect of HES1 on PP6. Whether PP6 has a regulatory effect on HES1 should be verified in PP6 CKO mice.

5.The authors identified SET, AURKA and IGBP1 as PP6-interactive proteins by mass spectrometry, and then only selected IGBP1 because it is regulated by L-menthol at the transcription level. Does L-menthol regulate the protein level of SET and AURKA? Does HES1 influence SET and AURKA? At the protein level, do SET and AURKA regulate Pp6? Although HES1 is a transcription factor, it can also regulate protein ubiquitination and phosphorylation.

6.Figure 6 demonstrates the regulatory relationship between Hes1, IGBP1 and Pp6. If the addition of L-menthol weakens the ubiquitination of Pp6 which consistent with the overexpression of IGBP1, and L-menthol simultaneously increases the expression of Hes1, IGBP1 and Pp6, the results will be more complete.In Figure 6d, compared with the control group, why is the protein level of Pp6 not down-regulated after silIGBP1?

7.There are several mistaken of the writing, please check the full text and figures carefully. For example, the "16.8nM" in line 117 should be "16.8 nM". Line 109, "upreguation" should be "upregulation", and the case format of "IGBP1" and "Hes1" should be unified in text and figures.

Reviewer #2 (Remarks to the Author):

The authors performed chemical library screening and found that L-menthol can up-regulate PP6, whose down-regulation leads to psoriasis. They analyzed the mechanism of how L-

menthol can up-regulate PP6 and showed that L-menthol directly interacts with HES1 protein and inhibits its degradation, and that HES1 is indispensable for PP6 up-regulation and suppression of psoriasis phenotypes. Further analyses demonstrated that HES1 transcriptionally increased Igbp1 expression and that IGBP1 increased PP6 protein by inhibiting its ubiquitination. From these data, the authors concluded that dysfunction of the HES1-IGBP1-PP6 axis is responsible for psoriasis pathology.

While the findings are interesting, the mechanistic part is obscure. Specific comments are indicated below.

1. In Fig. 5g, the increase of IGBP1 by HES1 is rather small, and it is not clear whether HES1 up-regulation is sufficient for suppression of psoriasis pathology. The authors should test this possibility.

2. HES1 is a well-known transcriptional repressor, and the mechanism of how HES1 up-regulates Igbp1 expression is not clear. It was previously shown that CaMK2d can switch HES1 from a transcriptional repressor to a transcriptional activator by phosphorylation of S126 (Ju et al. *Cell* 119:815-829, 2004; Sugita et al. *PNAS* 112:3080-3085, 2015). The authors should test this possibility.

3. In Fig. S1f, 100ug/day of L-menthol did not show clear effects on cell infiltration, although HES1 expression was increased (Fig. 3a). The authors should explain why.

4. In Fig. S2b, labeling for K5.Pp6fl/fl and WT could be wrong.

5. I am not sure about the origin and structure of Hes1 flox mice. The authors acknowledged XY Hu (Tsinghua University, Beijing, China), but his paper using the mice (Shang et al. (2016) *Nat. Immun.* 17, 930-937) acknowledged R. Kageyama (Kyoto University) for Hes1 flox mice by citing Imayoshi et al. (2008) *Development* 135, 2531-2541. The authors should correctly reference the paper for Hes1 flox mice.

Reviewer #3 (Remarks to the Author):

- The relationship between HES1 and psoriasis is not innovative and new. Several papers dealing with this topic have been published during the past decade (Wang et al., *Mediators Inflamm* 2020, doi: 10.1155/2020/8297134; Ma et al., *Mediators Inflamm* 2018; doi: 10.1155/2018/3069521; Kim et al., *Ann Dermatol.* 2016, doi: 10.5021/ad.2016.28.1.45; Yin et al., *Dermatology.* 2012, doi: 10.1159/000342359; Zhang et al., *Br J Dermatol.* 2010, doi: 10.1111/j.1365-2133.2010.09790.x.).

- HES1 is not the major key player in psoriasis. Therefore, targeting HES1 by menthol will not be a game changer and other strategies are more promising (Hawkes et al., *J Allergy Clin Immunol.* 2017 doi: 10.1016/j.jaci.2017.07.004; Ogawa et al., *J Dermatol.* 2018, doi: 10.1111/1346-8138.14139; Chuang et al., *Expert Opin Drug Discov.* 2018 Jun;13(6):551-562. doi: 10.1080/17460441.2018.1463214; Tokuyama and Mabuchi, *Int J Mol Sci.* 2020, doi: 10.3390/ijms21207488).

- Menthol as natural product has many biological activities and is therefore not specific in HES1 inhibition.

- HES1 is ubiquitously expressed in the body and very strongly expressed in smooth and skeletal muscles. HES1 is known as a negative regulator of myogenesis. This means that inhibition of HES1 may increase myogenesis. Indeed, HES1 expression has been correlated with rhabdomyosarcoma (Sang et al., Trends Mol Med 2010, 16: 17-26). Hence, high doses of menthol to inhibit HES1 may not only increase myogenesis but also increase the danger of rhabdomyosarcoma induction.

Reviewer #4 (Remarks to the Author):

In this manuscript, Wang et al. demonstrated that psoriasis developed through the dysfunctional HES1-IGBP1-PP6 axis, which L-menthol could reverse. This report paves the way for a novel therapeutic use of topical L-menthol for psoriasis. This paper is well-written with relevant data obtained from human psoriasis as well as mouse model. However, this reviewer has some concerns.

1. Fig. 1: This reviewer does not accept TLR7 expression in epidermal keratinocytes. Given that HaCaT cells were stimulated with R848, they should show the evidence of TLR7 expression in them by either immunostaining or Western blotting. Since HaCaT is an immortal cell line, and therefore, they have distinct proliferative and differential programs from normal keratinocytes, they should reproduce results of Fig. 1b using normal human keratinocytes.
2. Or they can use IL-17 stimulation instead of R848, as they previously shown in Immunity, 2020.
3. Fig. 1b: Control experiment in the absence of R848 is mandatory.
4. Fig. 1f, i, l: Imiquimod-untreated controls should be shown for the effect of L-menthol.
5. Fig. 3f: HES1 expression was seen in the basal layer of the epidermis in normal skin (particularly, left panel), although its expression was confined to KRT1, not to KRT5. Why?
6. Please show the PP6 distribution in normal and psoriatic skin, too.
7. Please show hyper-proliferation of normal keratinocyte cultures in vitro when knocked down of PP6 or HES gene.
8. Please show abnormal differentiation, such as loss of granular layer and hyperkeratosis, in the 3-D human skin equivalent culture when use of keratinocytes knocked down of PP6 or HES gene.
9. Before clinical use, it would be extremely relevant to show the therapeutic effect of L-menthol on the psoriasiform 3-D human skin equivalent developed with addition of cytokine mixtures, such as IFN-gamma, IL-17A and TNF.
10. They should confirm their previous knowledges of the role of PP6 for the consistency; for example, what is the role of mir31 or NF-kB pathway in the HES1-IGBP1-PP6 axis? Does L-menthol normalize the PP6 downstream pathway such as C/EBP-b and subsequent arginase-1 production?
11. Fig.S1 e, f: High concentration of L-menthol at 100µg did not work. Why? Was that because of the inhibition of nociception?
12. Is the dysfunctional HES1-IGBP1-PP6 axis specific in the psoriatic epidermis? They should examine other inflammatory skin diseases, like atopic dermatitis, lichen planus, or any NF-kB-mediated diseases, to see whether PP6 is downregulated in the epidermis.

Point-by-point response to the reviewers' comments

Reviewer 1

In this study, the authors identified that the natural compound L-menthol improves psoriatic skin inflammation by up-regulating Pp6 in keratinocytes, and revealed that L-menthol targets Hes1 and up-regulates the co-binding protein IGBP1 of Pp6, which inhibits the degradation of Pp6. This is a novel study that validates L-menthol as a potential drug for psoriasis. The whole experimental design is clear. However, some weaknesses in the data and its representation should be addressed.

1. **Regarding the design of animal experiment, the IMQ-induced psoriasis mice were treated with L-menthol at the same time of modeling for 7 days. Whether there is any influence on the modeling effect when using similar topical administration methods? In this case, is the role of drug intervention to protect or treat?**

Re: To further investigate the role of L-menthol in treating psoriasis, we first induced the skin inflammation in mice with IMQ for 5 days and then treated these mice with topical use of L-menthol (Revised Supplementary Fig. 4a). L-menthol relieved skin inflammation and upregulated Pp6 in the epidermis (Revised Supplementary Fig. 4b-g).

2. **In Supplementary Figure 1C, 20 μ M L-menthol seems to be less effective in stimulating PP6 than 10 μ M. Why? Figure S1 E and F show that different concentrations of L-menthol have completely different effect on acanthosis and dermal cell infiltration. Consequently, the following experiments showing that L-menthol relieves skin inflammation should specify the specific concentration of L-menthol.**

Re: (1) We used CCK-8 to detect the cell viability of inflamed HaCaT cells treated with L-menthol. We found that 20 μ M L-menthol decreased the cell viability of inflamed HaCaT cells, while 10 μ M L-menthol did not (Revised Supplementary Fig. 2d). The effect of L-menthol on cell viability may account for that 20 μ M L-menthol seems to be less effective in stimulating PP6 than 10 μ M L-menthol.

(2) In the following experiments, 50 μ g/day L-menthol relieves IMQ-induced skin inflammation. We specify the concentration of L-menthol in the figure legends.

3. **Figures 3A, 3B and S3C verify the interaction of HES1 and L-menthol in HaCaT cells, primary mouse keratinocytes and purified HES1 protein. It should also be better validated in the keratinocytes of mice with psoriasis. In addition to regulating the proteasome degradation of HES1 (figure 3I), does L-menthol affect the protein translation**

process of HES1? CHX tracking experiment should be designed here. Does L-menthol affect the protein level of HES1 ubiquitination?

Re: (1) To verify the interaction between Hes1 and L-menthol in the keratinocytes of mice with psoriasis, we took advantage of an unbiased target identification approach, DARTS. The epidermis of mice with IMQ induction was used as the protein source for the DARTS assay. The interaction of HES1 and L-menthol was verified in the epidermis of mice with IMQ induction (Revised Supplementary Fig. 7d).

(2) To perform the cycloheximide (CHX) tracking experiment, we treated HaCaT cells with 1 µg/ml R848 and DMSO or 10 µM L-menthol along with 25 µg/ml cycloheximide (CHX) for the indicated time. L-menthol prolonged the half-life of HES1 in the presence of CHX (Revised Supplementary Fig. 7f), indicating that L-menthol protected the already translated HES1 from degradation.

(3) To investigate whether L-menthol affects HES1 ubiquitination, we immunoblotted HES1-immunoprecipitated proteins in HaCaT cells treated with DMSO (Ctr) or 10 µM L-menthol for 12 hours in the presence of 1 µg/ml R848. L-menthol led to a decrease of ubiquitinated HES1 in inflamed keratinocytes (Revised Supplementary Fig. 7g)

4. **Regarding the regulation study between HES1 and PP6, Figure 4 demonstrate the up-regulation effect of HES1 on PP6. Whether PP6 has a regulatory effect on HES1 should be verified in PP6 CKO mice.**

Re: To study whether PP6 has a regulatory effect on HES1, we used *Pp6^{fl/fl}* mouse and *K5.Pp6^{fl/fl}* mouse (PP6 conditional knock out mice). The deletion of *Pp6* did not affect the protein level of Hes1 in the epidermis, indicating that PP6 did not have a regulatory effect on HES1 (see below).

5. The authors identified SET, AURKA, and IGBP1 as PP6-interactive proteins by mass spectrometry, and then only selected IGBP1 because it is regulated by L-menthol at the transcription level. Does L-menthol regulate the protein level of SET and AURKA? Does HES1 influence SET and AURKA? At the protein level, do SET and AURKA regulate Pp6? Although HES1 is a transcription factor, it can also regulate protein ubiquitination and phosphorylation.

Re: (1) To evaluate whether L-menthol regulates the protein level of SET and AURKA, we treated HaCaT cells with 10 μ M L-menthol in the presence of 1 μ g/ml R848. The immunoblotting result showed that L-menthol did not affect the protein level of SET or AURKA in inflamed keratinocytes (Revised Supplementary Fig. 9b)

(2) To test whether HES1 influences SET and AURKA, we transfected HaCaT cells with scramble siRNA (Ctr) or with HES1 siRNA (si-HES1). Cell lysates were immunoblotted with anti-HES1, anti-SET, anti-AURKA, anti-Tubulin, or anti- β -actin. Downregulation of HES1 in keratinocytes did not affect the protein level of SET or AURKA (see below).

(3) To investigate whether SET and AURKA regulate PP6, we transfected HaCaT cells with scramble siRNA (Ctr) or with SET siRNA (si-SET) or with AURKA siRNA (si-AURKA). Cell lysates were immunoblotted with anti-PP6, anti-SET, anti-AURKA, anti-Tubulin or anti- β -actin. At the protein level, down-regulation of SET led to the decrease of PP6 in keratinocytes, while down-regulation of AURKA did not affect PP6 (see below).

6. Figure 6 demonstrates the regulatory relationship between Hes1, IGBP1 and Pp6. If the addition of L-menthol weakens the ubiquitination of Pp6 which consistent with the overexpression of IGBP1, and L-menthol simultaneously increases the expression of Hes1, IGBP1 and Pp6, the results will be more complete. In Figure 6d, compared with the control group, why is the protein level of Pp6 not down-regulated after siIGBP1?

Re: (1) To study whether L-menthol affects the ubiquitination of Pp6, we immunoblotted PP6-immunoprecipitated ubiquitin in HaCaT cells treated with DMSO (Ctr) or 10 μ M L-menthol for 12 hours in the presence of 1 μ g/ml R848. L-menthol weakens the ubiquitination of PP6 in inflamed keratinocytes, which is consistent with the effect of IGBP1 overexpression (Revised Supplementary Fig. 11c).

(2) In previous Figure 6d, we adjusted the loading volume to make sure that PP6 in the immunoprecipitates pulled by its antibody was similar between the two groups. Thus, the increase of PP6-ubiquitination caused by IGBP1 downregulation was easy to be observed. Before we got the immunoprecipitates pulled by the PP6 antibody, the protein level of PP6 was down-regulated by si-IGBP1 in the cell lysates (Revised Fig. 6c).

7. There are several mistaken of the writing, please check the full text and figures carefully. For example, the “16.8nM” in line 117 should be “16.8 nM”. Line 109, “upreguation” should be “upregulation”, and the case format of “IGBP1” and “Hes1” should be unified in text and figures.

Re: We are very sorry for the mistakes in this manuscript and the inconvenience they caused in reading. The manuscript has been thoroughly revised. The case formats of “IGBP1” and

“Hes1” in text and figures were different because the samples were derived from either mice or humans. We used “Pp6”, “Hes1”, and “Igbp1” to describe proteins from mice, and we used “PP6”, “HES1”, and “IGBP1” to describe proteins from humans. Thank you for your useful comments.

Reviewer 2

The authors performed chemical library screening and found that L-menthol can up-regulate PP6, whose down-regulation leads to psoriasis. They analyzed the mechanism of how L-menthol can up-regulate PP6 and showed that L-menthol directly interacts with HES1 protein and inhibits its degradation, and that HES1 is indispensable for PP6 up-regulation and suppression of psoriasis phenotypes. Further analyses demonstrated that HES1 transcriptionally increased Igbp1 expression and that IGBP1 increased PP6 protein by inhibiting its ubiquitination. From these data, the authors concluded that dysfunction of the HES1-IGBP1-PP6 axis is responsible for psoriasis pathology.

While the findings are interesting, the mechanistic part is obscure. Specific comments are indicated below.

- 1. In Fig. 5g, the increase of IGBP1 by HES1 is rather small, and it is not clear whether HES1 up-regulation is sufficient for suppression of psoriasis pathology. The authors should test this possibility.**

Re: To investigate whether HES1 up-regulation suppressed psoriasis pathology, we injected a Hes1-expressing adeno-associated virus serotype 9 (AAV-Hes1) intracutaneously to increase Hes1 levels in mice treated with IMQ (Revised Supplementary Fig. 10a-b). The epidermal thickness, dermal cellular infiltrates and epidermal Ki67⁺ cell counts in skin lesions were significantly decreased in IMQ-induced mice given AAV-Hes1 compared with mice administered AAV-Control (Revised Supplementary Fig. 10c-g). Meanwhile, the protein levels of Hes1, Igbp1 and Pp6 were increased in the epidermis of IMQ-induced mice given AAV-Hes1 (Revised Supplementary Fig. 10h). Thus, HES1 up-regulation remarkably suppressed the psoriasis-like skin inflammation in mice.

- 2. HES1 is a well-known transcriptional repressor, and the mechanism of how HES1 up-regulates Igbp1 expression is not clear. It was previously shown that CaMK2d can switch HES1 from a transcriptional repressor to a transcriptional activator by phosphorylation of S126 (Ju et al. Cell 119:815-829, 2004; Sugita et al. PNAS 112:3080-3085, 2015). The authors should test this possibility.**

Re: To explore the mechanism of how HES1 promotes the transcription of IGBP1, we mutated the serine residue of the CaMK2d phosphorylation site in HES1 to inhibit Hes1-CaMK2d

cooperation. Compared with the wild-type HES1, the kinase domain mutated (S126A) form of HES1 did not alter the mRNA level of IGBP1, indicating that the phosphorylation of S126 in HES1 is dispensable for the activation of IGBP1 in keratinocytes (Revised Supplementary Fig. 9c-d).

- 3. In Fig. S1f, 100ug/day of L-menthol did not show clear effects on cell infiltration, although HES1 expression was increased (Fig. 3a). The authors should explain why.**

Re: (1) In previous Fig. 3a, we used drug affinity responsive target stability assay (DARTS) to identify the target of L-menthol in keratinocytes. HaCaT cell lysates were first incubated with L-menthol and then subjected to pronase digestion. HES1 was upregulated by L-menthol, indicating that L-menthol targets HES1 and protects it from being digested by pronase.

(2) In previous Fig. S1f, 100 µg/day of L-menthol did not show clear effects on cell infiltration, indicating that high concentration may affect the effect of L-menthol. To further investigate the mechanism, we used CCK-8 to detect the cell viability of inflamed HaCaT cells treated with L-menthol. We found that 20 µM L-menthol decreased the cell viability of inflamed HaCaT cells, while 10 µM L-menthol did not (Revised Supplementary Fig. 2d). Thus, the high concentration of L-menthol decreases the cell viability, which may account for the phenomenon.

- 4. In Fig. S2b, labeling for K5. Pp6^{n/n} and WT could be wrong.**

Re: We are very sorry for the mistakes in this manuscript. In Revised Supplementary Fig. 6b, we have corrected the mistake. The red arrow shows the K5-Cre-mediated epidermal-specific deletion of Pp6.

- 5. I am not sure about the origin and structure of Hes1 flox mice. The authors acknowledged XY Hu (Tsinghua University, Beijing, China), but his paper using the mice (Shang et al. (2016) Nat. Immun. 17, 930-937) acknowledged R. Kageyama (Kyoto University) for Hes1 flox mice by citing Imayoshi et al. (2008) Development 135, 2531-2541. The authors should correctly reference the paper for Hes1 flox mice.**

Re: We are very sorry for the mistakes in this manuscript and the inconvenience they caused in reading. We have correctly referenced the paper for *Hes1*^{n/n} mice by citing Imayoshi et al. (2008) Development 135, 2531-2541. Thank you for your helpful comments.

Reviewer: 3

- 1. The relationship between HES1 and psoriasis is not innovative and new. Several papers dealing with this topic have been published during the past decade (Wang et al., *Mediators Inflamm* 2020, doi: 10.1155/2020/8297134; Ma et a., *Mediators Inflamm* 2018; doi: 10.1155/2018/3069521; Kim et al., *Ann Dermatol.* 2016, doi: 10.5021/ad.2016.28.1.45; Yin et al., *Dermatology.* 2012, doi: 10.1159/000342359; Zhang et al., *Br J Dermatol.* 2010, doi: 10.1111/j.1365-2133.2010.09790.x.).**

Re: Thank you for your comment.

The abnormal expression of HES1 in psoriasis has been reported, but the exact function of HES1 in keratinocytes was not clarified using genetic approaches. We here found that keratinocyte-specific Hes1 ablation exacerbated psoriasis-like skin inflammation in mice (Revised Supplementary Fig. 8a-f), while HES1 upregulation relieves psoriasis-like skin inflammation (Revised Supplementary Fig. 10a-h). HES1 was the direct target of L-menthol in keratinocytes (Revised Fig. 3a-c, Revised Supplementary Fig. 7a-d), which governed PP6-upregulating function of L-menthol in treating psoriasis-like skin inflammation (Revised Fig. 4a-h). Mechanistically, HES1 transcriptionally activated the expression of IGBP1 (Revised Fig. 5e-g) which promotes PP6 expression and inhibits its ubiquitination (Revised Fig. 6e-f). Our study is the first to reveal the role of HES1-IGBP1-PP6 axis in psoriasis.

- 2. HES1 is not the major key player in psoriasis. Therefore, tageting HES1 by menthol will not be a game changer and other strategies are more promising (Hawkes et al., *J Allergy Clin Immunol.* 2017 doi: 10.1016/j.jaci.2017.07.004; Ogawa et a., *J Dermatol.* 2018, doi: 10.1111/1346-8138.14139; Chuang et a., *Expert Opin Drug Discov.* 2018 Jun;13(6):551-562. doi: 10.1080/17460441.2018.1463214; Tokuyama and Mabuchi, *Int J Mol Sci.* 2020, doi: 10.3390/ijms21207488).**

Re: Thank you for your comment.

(1) Our data showed that HES1 is decreased in lesional psoriatic skin (Revised Fig. 3d-f) and HES1 deficiency in keratinocytes exacerbates psoriasis-like skin inflammation (Revised Supplementary Fig. 8a-f). We also found that HES1 upregulation relieves psoriasis-like skin inflammation (Revised Supplementary Fig. 10a-h). All of the data elucidate the important role of HES1 in psoriasis.

(2) Emerging bispecific antibodies offer the potential for even better disease control, whereas small-molecule drugs offer future alternatives to the use of biologics and less costly long-term disease management (Hawkes et al., *J Allergy Clin Immunol.* 2017, 140(3):645-653).

(3) In our study, L-menthol targets HES1 in keratinocytes (Revised Fig. 3a-c, Revised Supplementary Fig. 7a-d) and relieves IMQ-induced skin inflammation (Revised Fig. 1c-l, Revised Supplementary Fig. 4a-g). Furthermore, L-menthol decreased the epidermal thickness

in psoriasiform 3-D human skin equivalents (Revised Supplementary Fig. 5b-e). Together, targeting HES1 by L-menthol will be promising in the treatment of psoriasis.

3. Menthol as a natural product has many biological activities and is therefore not specific in HES1 inhibition.

Re: Thank you for your comment.

Through target identification approach, we found that HES1 is among the most abundant and enriched proteins present in the L-menthol treated sample (Revised Supplementary Fig. 7b). Meanwhile, L-menthol targets HES1 in keratinocytes (Revised Fig. 3a-c, Revised Supplementary Fig. 7a-d). Besides, L-menthol upregulates HES1 in inflamed keratinocytes by inhibiting its proteasomal degradation (Revised Fig. 3g-i). Furthermore, by using Hes1 conditional knock-out mice in keratinocytes, we demonstrated that HES1 is essential for the therapeutic effect of L-menthol (Revised Fig. 4a-h). Thus, L-menthol targets and upregulates HES1 in inflamed keratinocytes, and ameliorates psoriasis-like skin inflammation.

4. HES1 is ubiquitously expressed in the body and very strongly expressed in smooth and skeletal muscles. HES1 is known as a negative regulator of myogenesis. This means that inhibition of HES1 may increase myogenesis. Indeed, HES1 expression has been correlated with rhabdomyosarcoma (Sang et al., Trends Mol Med 2010, 16: 17-26). Hence, high doses of menthol to inhibit HES1 may not only increase myogenesis but also increase the danger of rhabdomyosarcoma induction.

Re: Thank you for your comment.

(1) The expression pattern of HES1 is different from rhabdomyosarcomas and psoriasis. Our data showed that HES1 is mainly expressed by keratinocytes in the epidermis and dramatically decreased in lesional psoriatic skin (Revised Fig. 3d-f). L-menthol upregulates HES1 in inflamed keratinocytes by inhibiting its proteasomal degradation (Revised Fig. 3g-i) and ameliorates psoriasis-like skin inflammation (Revised Fig. 1c-l, Revised Supplementary Fig. 4a-g).

(2) Topical application of L-menthol minimizes a systemic influence, which does not affect skin homeostasis under the normal condition (Revised Supplementary Fig. 3a-f). Thus, L-menthol is a safe treatment for psoriasis.

Reviewer 4

In this manuscript, Wang et al. demonstrated that psoriasis developed through the dysfunctional HES1-IGBP1-PP6 axis, which L-menthol could reverse. This report paves the way for a novel therapeutic use of topical L-menthol for psoriasis. This paper is well-written with relevant data obtained from human psoriasis as well as mouse model. However, this reviewer has some concerns.

1. **Fig. 1: This reviewer does not accept TLR7 expression in epidermal keratinocytes. Given that HaCaT cells were stimulated with R848, they should show the evidence of TLR7 expression in them by either immunostaining or Western blotting. Since HaCaT is an immortal cell line, and therefore, they have distinct proliferative and differential programs from normal keratinocytes, they should reproduce results of Fig. 1b using normal human keratinocytes.**

Re: (1) By using western blotting and THP-1 cells as the positive control, we showed the evidence of TLR7 expression in HaCaT cells (Revised Supplementary Fig. 2b). Besides, it has been reported that TLR7 is expressed in HaCaT cells (Jia B et al., Mol Biol Rep. 2013, 40(11):6363-6369. doi:10.1007/s11033-013-2750-9).

(2) We treated IL-17A-stimulated primary normal human epidermal keratinocytes (NHEK) with L-menthol. PP6 was decreased in IL-17A-stimulated NHEK cells compared to that in vehicle-treated cells, and L-menthol treatment rescued the expression of PP6 in IL-17A-stimulated NHEK cells (Supplementary Fig. 5a).

2. **Or they can use IL-17 stimulation instead of R848, as they previously shown in Immunity, 2020.**

Re: L-menthol treatment rescued the expression of PP6 in IL-17A-stimulated NHEK cells (Supplementary Fig. 5a).

3. **Fig. 1b: Control experiment in the absence of R848 is mandatory.**

Re: Control experiment in the absence of R848 was shown in the revised Fig. 1b. PP6 was decreased in R848-stimulated HaCaT cells compared to that in vehicle-treated cells, and L-menthol treatment rescued the expression of PP6 in R848-stimulated HaCaT cells.

4. **Fig. 1f, i, l: Imiquimod-untreated controls should be shown for the effect of L-menthol.**

Re: Imiquimod-untreated controls were shown in revised Supplementary Fig. 3a-g. L-menthol did not affect the skin homeostasis in the mice without IMQ induction.

5. **Fig. 3f: HES1 expression was seen in the basal layer of the epidermis in normal skin (particularly, left panel), although its expression was confined to KRT1, not to KRT5. Why?**

Re: (1) Immunohistochemical analyses showed HES1 expression was seen both in the basal layer and the spinous layer of the epidermis in normal skin. HES1 was dramatically decreased in the epidermis of psoriasis patients compared with that of healthy controls. Below is the immunohistochemical analysis of HES1.

(2) We performed single-cell RNA sequencing (scRNA-seq) on the epidermis derived from healthy donors and psoriasis patients (Revised Fig. 3d). HES1 was expressed in Keratin 1 (K1)-positive differentiated keratinocytes and Keratin 5 (K5)-positive differentiated keratinocytes. However, the ratio of HES1^{hi} keratinocytes in K1 positive cells was higher than that in K5 positive cells in the epidermis of healthy controls. Below is the ratio of HES1⁺ cells.

6. Please show the PP6 distribution in normal and psoriatic skin, too.

Re: We showed the PP6 distribution in normal and psoriatic skin in revised Supplementary Fig. 1a. Immunohistochemical analyses showed that PP6 was dramatically decreased in the

epidermis of psoriasis patients compared with that of healthy controls (Revised Supplementary Fig. 1a).

7. Please show hyper-proliferation of normal keratinocyte cultures in vitro when knocked down of PP6 or HES gene.

Re: (1) We transfected NHEK cells with scramble siRNA (Ctr) or with HES1 siRNA (si-HES1). Cell lysates were immunoblotted with anti-HES1 or anti- β -actin. Cell viability was measured by CCK-8. We found that HES1 down-regulation in NHEK cells led to the hyper-proliferation of normal keratinocyte cultures (Revised Supplementary Fig. 8l-m).

(2) The epidermal Ki67⁺ cell counts in skin lesions derived from IMQ-treated *K5. Hes1^{fl/fl}* mice were significantly augmented compared with IMQ-treated *Hes1^{fl/fl}* mice (Revised Supplementary Fig. 8e-f), indicating that *Hes1* deletion leads to the hyper-proliferation of keratinocytes.

8. Please show abnormal differentiation, such as loss of granular layer and hyperkeratosis, in the 3-D human skin equivalent culture when use of keratinocytes knocked down of PP6 or HES gene.

Re: (1) We established the 3-D human skin equivalent culture with keratinocytes knocked down of *PP6* gene. Increased epidermal thickness was observed in the 3-D human skin equivalent culture with keratinocytes knocked down of *PP6* gene (Supplementary Fig. 1b-f), indicating that PP6 deficiency in keratinocytes promotes epidermal hyperplasia.

(2) Previous studies showed that epidermis-specific Pp6-deficient mice spontaneously develop psoriasis-like skin inflammation. Histological examination of the skin lesions showed epidermal hyperplasia (acanthosis) with loss of the granular layer, hyperkeratosis, and parakeratosis in the epidermis together with microabscesses that accumulated on the surface of the thickened epidermis and massive cellular infiltrates in the dermis (Lou F et al., *Immunity*. 2020, 53(1):204-216.e10.).

(3) Below is the H&E staining of the skin lesions from the upper back of a *K5. Pp6^{fl/fl}* mouse (Lou F et al., *Immunity*. 2020, 53(1):204-216.e10.). a, acanthosis; b, increased proliferative basal layer epidermal keratinocytes; d, dermal cell infiltrates; k, hyperkeratosis; p, parakeratosis; g, loss of stratum granulosum; m, Munro's microabscess. The dotted line indicates the border between the epidermis and the dermis; scale bar, 100 μ m.

9. Before clinical use, it would be extremely relevant to show the therapeutic effect of L-menthol on the psoriasiform 3-D human skin equivalent developed with addition of cytokine mixtures, such as IFN-gamma, IL-17A and TNF.

Re: 3-D human skin equivalent was stimulated with a combination of 20 ng/ml IL-22 and 5 ng/ml TNF- α (Wolk K et al., J Mol Med (Berl). 2009, 87(5):523-536. doi:10.1007/s00109-009-0457-0). After 24 h, psoriasiform 3-D human skin equivalents were treated with DMSO or 10 μ M L-menthol. After 48 h, 3-D skin cultures were processed for immunohistochemical analyses and western blot. We found that L-menthol decreased the epidermal thickness and upregulated PP6 in the psoriasiform 3-D human skin equivalent (Revised Supplementary Fig. 5b-e).

10. They should confirm their previous knowledges of the role of PP6 for the consistency; for example, what is the role of mir31 or NF-kB pathway in the HES1-IGBP1-PP6 axis? Does L-menthol normalize the PP6 downstream pathway such as C/EBP-b and subsequent arginase-1 production?

Re: (1) To study the role of miR-31 in the Hes1-Igfbp1-Pp6 axis, we treated miR-31^{-/-} mice (Zhou H et al., EMBO Rep. 2022;23(5): e53475.) and WT mice with IMQ for 7 days and isolated their epidermis for qPCR detection and immunoblotting analysis. We found that miR-31 deletion leads to the increase of Hes1, Igfbp1, and Pp6 in the epidermis of IMQ-induced mice. Below is the qPCR detection and immunoblotting analysis.

(2) L-menthol decreased the phosphorylation of C/EBP- β and down-regulated ARG1 in inflamed keratinocytes (Revised Supplementary Fig. 2i-j), indicating that L-menthol normalized the PP6 downstream pathway.

11. **Fig.S1 e, f: High concentration of L-menthol at 100 μ g did not work. Why? Was that because of the inhibition of nociception?**

Re: To further investigate the mechanism, we used CCK-8 to detect the cell viability of inflamed HaCaT cells treated with L-menthol. We found that 20 μ M L-menthol decreased the cell viability of inflamed HaCaT cells, while 10 μ M L-menthol did not (Revised Supplementary Fig. 2d). Thus, the high concentration of L-menthol decreases the cell viability, which may account for the phenomenon.

12. **Is the dysfunctional HES1-IGBP1-PP6 axis specific in the psoriatic epidermis? They should examine other inflammatory skin diseases, like atopic dermatitis, lichen planus, or any NF- κ B-mediated diseases, to see whether PP6 is downregulated in the epidermis.**

Re: We examined the expression of PP6 in Vitiligo which is an NF- κ B-mediated inflammatory skin disease (Becatti M et al., Antioxid Redox Signal. 2010;13(9):1309-1321.) and found that there is no significant change in PP6 expression between the Vitiligo skin and normal skin.

Below is the immunohistochemical analyses of PP6 in the epidermis of Vitiligo patients and healthy donor.

Normal skin

Vitiligo skin

REVIEWER COMMENTS

Reviewer #1 (Remarks to the Author):

The authors have sufficiently addressed my comments. The manuscript is significantly improved compared to the original submission.

Reviewer #2 (Remarks to the Author):

The authors addressed my concerns properly, and now I support the acceptance of the paper.

Reviewer #3 (Remarks to the Author):

The authors responded adequately to the critical points raised.

Reviewer #4 (Remarks to the Author):

The revised manuscript by Wang et al. has been substantially improved by additional data in response to reviewers' comments. However, there are still some concerns.

1. Sup FigS5: They presented the data showing that IL-17 reduce pp6 level in NHEK and L-menthol reversed it. However, the psoriasiform 3-D skin equivalent culture was prepared under stimulation with mixed cytokines of IL-22 and TNF. Why not IL-17? Did the combination of these two cytokines also affect pp6?

2. Please show data of reduction of Hes1 protein in NHEK by IL-17 stimulation.

3. Please reproduce the same results of Figure 3h,i using NHEK under stimulation with IL-17.

4. What is the underlying mechanism of IL-17 for the HES1-axis.

5. Vitiligo is not a relevant control skin disease since the epidermis was not hyperplastic. Please check out PP6 or Hes1 in samples from chronic eczema like atopic dermatitis showing acanthosis which is reportedly induced by IL-17 and/or IL-22.

6. Fig S8m: They showed an increased viability of siHES1-introduced NHEK cells. The description that Hes1 downregulation led to the hyper-proliferation of NHEK cells was incorrect. They should assess cell proliferation using MTT assay, for example.

7. Their method to assess the epidermal thickness for hyperplasia or acanthosis is incorrect. It should be shown by the length from the basal membrane to the cornified layer of the epidermis, not by area. They should amend them all.

Point-by-point response to the reviewers' comments

Reviewer 4

The revised manuscript by Wang et al. has been substantially improved by additional data in response to reviewers' comments. However, there are still some concerns.

1. **Sup FigS5: They presented the data showing that IL-17 reduce pp6 level in NHEK and L-menthol reversed it. However, the psoriasiform 3-D skin equivalent culture was prepared under stimulation with mixed cytokines of IL-22 and TNF. Why not IL-17? Did the combination of these two cytokines also affect pp6?**

Re: (1) Previous studies showed that IL-22 but not IL-17 or IFN- γ induced acanthosis of cultures and decreased granularity in the upper living layer. Below is the histological analysis of human epidermis model stimulated with IL-22, IFN- γ , and IL-17 for 72 h. (a, b) Tissues were analyzed by histology (hemalaun). Images represent the result of one out of three independent experiments for each experimental series. Microscopic magnification, 100-fold. (c, d) Mean (\pm SEM) thickness data of living cell layers from three independent experiments described in (a) and (b) are given as percent of control (Wolk K et al., J Mol Med (Berl). 2009, 87(5):523-536. doi:10.1007/s00109-009-0457-0).

(2) Wolk K et al found that TNF- α led to an only very weak if any increase in epidermal thickness with some spongiosis, but, in combination with IL-22, provoked a more pronounced epidermal thickness than did IL-22 alone. Below is the histological analysis (hemalaun) of

human epidermis model stimulated with IL-22, TNF- α , a combination of IL-22 and TNF- α , or without stimulation (control) for 72 h. Images represent the result from two experiments. Microscopic magnification, 100-fold (Wolk K et al., J Mol Med (Berl). 2009, 87(5):523-536. doi:10.1007/s00109-009-0457-0).

(3) Thus, to obtain the human skin model with increased epidermal thickness, the psoriasiform 3-D skin equivalent culture was prepared under stimulation with mixed cytokines of IL-22 and TNF, but not IL-17.

(4) To investigate the effect of the combination of IL-22 and TNF- α on PP6, we stimulated 3-D human skin equivalent with or without a combination of 20 ng/ml IL-22 and 5 ng/ml TNF- α for 72h. Cell lysates were immunoblotted with anti-PP6 or anti- β -actin. The combination of IL-22 and TNF- α downregulated PP6 in 3-D human skin equivalent (see below).

2. Please show data of reduction of Hes1 protein in NHEK by IL-17 stimulation.

Re: To investigate whether IL-17A reduced HES1 in primary normal human epidermal keratinocytes (NHEK), we treated NHEK cells with or without 200 ng/ml IL-17A for 12h. Cell lysates were immunoblotted with anti-HES1 or anti- β -actin. IL-17A reduced HES1 in NHEK cells (see below).

3. Please reproduce the same results of Figure 3h, i using NHEK under stimulation with IL-17.

Re: (1) To explore the effect of L-menthol on HES1 in IL-17A-stimulated NHEK cells, we treated IL-17A-stimulated NHEK cells with 10 μ M L-menthol for 12h. Cell lysates were immunoblotted with anti-HES1 or anti- β -actin. IL-17A reduced HES1 in NHEK cells, while L-menthol rescued HES1 in IL-17A-stimulated NHEK cells (see below).

(2) To explore whether L-menthol increased HES1 through proteasomal degradation pathway, we treated IL-17A-stimulated NHEK cells with 10 μ M L-menthol for 12h in the presence or absence of 0.5 μ M MG132. Cell lysates were immunoblotted with anti-HES1 or anti- β -actin. In the absence of MG132, L-menthol increased HES1 in IL-17A-stimulated NHEK cells. But in the presence of MG132, L-menthol failed to increase HES1 in IL-17A-stimulated NHEK cells (see below).

4. What is the underlying mechanism of IL-17 for the HES1-axis.

Re: Previous study showed that IL-17A activated the NF- κ B signaling pathway mediating miR-31 expression in keratinocytes (Yan S et al. Nat Commun. 2015; 6:7652.). To study the role of miR-31 in the Hes1-Igfbp1-Pp6 axis, we treated miR-31^{-/-} mice (Zhou H et al., EMBO Rep. 2022;23(5): e53475.) and WT mice with IMQ for 7 days and isolated their epidermis for qPCR detection and immunoblotting analysis. We found that miR-31 deletion led to the increase of Hes1, Igfbp1, and Pp6 in the epidermis of IMQ-induced mice (see below). Together, IL-17A induces miR-31 mediating the downregulation of HES1 in keratinocytes.

5. Vitiligo is not a relevant control skin disease since the epidermis was not hyperplastic. Please check out PP6 or Hes1 in samples from chronic eczema like atopic dermatitis showing acanthosis which is reportedly induced by IL-17 and/or IL-22.

Re: We performed immunofluorescent labeling of PP6 in the skin from healthy donors and patients with atopic dermatitis and found that PP6 was decreased in the epidermis of atopic dermatitis patients (see below).

6. **Fig S8m: They showed an increased viability of siHES1-introduced NHEK cells. The description that Hes1 downregulation led to the hyper-proliferation of NHEK cells was incorrect. They should assess cell proliferation using MTT assay, for example.**

Re: We are very sorry for the mistake in this manuscript. To investigate the role of HES1 in human keratinocytes, NHEK cells were infected with the adenovirus to mediate the downregulation of HES1. After 2 days, NHEK cells were subjected to western blot, or MTT cell proliferation assay (Beyotime cat. C0009S). Cell lysates were immunoblotted with anti-HES1 or anti- β -actin. Cell proliferation was measured by MTT assay. We found that HES1 down-regulation in NHEK cells led to the hyper-proliferation of normal keratinocyte cultures (Revised Supplementary Fig. 8l-m).

7. **Their method to assess the epidermal thickness for hyperplasia or acanthosis is incorrect. It should be shown by the length from the basal membrane to the cornified layer of the epidermis, not by area. They should amend them all.**

Re: We are very sorry for the mistake in this manuscript. Epidermal hyperplasia (acanthosis) was assessed by using average length of three times of measures from the basal membrane to the cornified layer of the epidermis. We have amended them all. Thank you for your helpful suggestions.

REVIEWERS' COMMENTS

Reviewer #4 (Remarks to the Author):

The revised manuscript is now acceptable.